# Identification and characterization of endo-α-, exo-α-, and exo-β-D-arabinofuranosidases degrading lipoarabinomannan and arabinogalactan of mycobacteria

Michiko Shimokawa[1,10], Akihiro Ishiwata [2,10], Toma Kashima[3,10], Chiho Nakashima[3], Jiaman Li[3], Riku Fukushima[3], Naomi Sawai[1], Miku Nakamori[1], Yuuki Tanaka[1], Azusa Kudo[1], Sae Morikami[1], Nao Iwanaga[1], Genki Akai[3], Nobutaka Shimizu [4], Takatoshi Arakawa [5], Chihaya Yamada[6], Kanefumi Kitahara[1], Katsunori Tanaka[2,7], Yukishige Ito[2,8], Shinya Fushinobu [3,9] ✉ & Kiyotaka Fujita [1] ✉

The cell walls of pathogenic and acidophilic bacteria, such as *Mycobacterium tuberculosis* and *Mycobacterium leprae*, contain lipoarabinomannan and arabinogalactan. These components are composed of D-arabinose, the enantiomer of the typical L-arabinose found in plants. The unique glycan structures of mycobacteria contribute to their ability to evade mammalian immune responses. In this study, we identified four enzymes (two GH183 endo-D-arabinanases, GH172 exo-α-D-arabinofuranosidase, and GH116 exo-β-D-arabinofuranosidase) from *Microbacterium arabinogalactanolyticum*. These enzymes completely degraded the complex D-arabinan core structure of lipoarabinomannan and arabinogalactan in a concerted manner. Furthermore, through biochemical characterization using synthetic substrates and X-ray crystallography, we elucidated the mechanisms of substrate recognition and anomer-retaining hydrolysis for the α- and β-D-arabinofuranosidic bonds in both endo- and exo-mode reactions. The discovery of these D-arabinan-degrading enzymes, along with the understanding of their structural basis for substrate specificity, provides valuable resources for investigating the intricate glycan architecture of mycobacterial cell wall polysaccharides and their contribution to pathogenicity.

Arabinose is distinct among the monosaccharides because it is found in nature in both the D- and L-enantiomers. L-Arabinose is a major component of plant cell walls as a component of pectin and is biosynthesized by pathways via UDP-D-glucose[1]. However, the biosynthetic pathway of D-arabinose is unique and has been identified only in several bacteria[2–5]. D-Arabinofuranosides (Ara*f*s) are one of the predominant cell wall components of actinomycetes, such as *Mycobacterium*, *Rhodococcus*, and *Nocardia* species[2]. Regarding immune modulation and pathogenesis in humans, researchers have been studying the biosynthetic pathways and carbohydrate structures of the cell wall components of lipoarabinomannan (LAM) and arabinogalactan (AG) in acid-fast bacteria, including *Mycobacterium tuberculosis*, *Mycobacterium leprae* (the causative pathogen of Hansen's disease or leprosy), and *Mycolicibacterium smegmatis* (formerly

*Mycobacterium smegmatis*[6–11]. While we have successfully synthesized the docosasaccharide D-arabinan motif[12], the complex D-arabinan structures of LAM and AG, which comprise multiple branches and both α- and β-D-Ara*f* bonds, have been challenging targets for total organic synthesis[13–15].

Glycoside hydrolases (GH) that cleave α-L-Ara*f* bonds have been extensively studied, and many families have been classified in the Carbohydrate-Active enZyme (CAZy) database[16], including GH43, GH51, GH54, and GH62. However, among the classified GH families until recently, only the α-D-fructofuranosidase (Fru*f*-ase) of GH172 hydrolyzes the α-D-Ara*f* bond, a recently reported side reaction for a similar glycosidic bond[17]. A bacterial endo-D-arabinanase that degrades mycobacterial LAM and AG was initially reported in the early 1970s[18–20], and similar activity was later reported in *Cellulomonas*[21,22] and *My. smegmatis*[23,24]. However, an endo-D-arabinanase-encoding gene has not yet been identified. Recently, exo- and endo-D-arabinanases were identified from *Dysgonomonas gadei*[25].

The extracellular endo-D-arabinanase producer[18], *Aureobacterium* sp. M-2 strain isolated by Kotani et al. is currently named *Microbacterium arabinogalactanolyticum* JCM 9171[T] (ref. [26]). Here, we report the identification of the genes encoding enzymes involved in mycobacterial LAM and AG degradation, and that these enzymes degrade the D-arabinan region cooperatively and completely. Our analysis of endo-D-arabinases from *Mi. arabinogalactanolyticum* revealed that they belonged to a new GH family (GH183). Among the neighboring genes, we found an exo-α-D-arabinofuranosidase (exo-α-D-Ara*f*-ase) belonging to GH172 and an exo-β-D-arabinofuranosidase (exo-β-D-Ara*f*-ase) belonging to GH116. We examined the substrate specificity of these enzymes using mycobacterial LAM and AG and synthesized oligo-D-arabinofuranoside substrates. Furthermore, we elucidated the reaction processes and substrate specificities of endo-D-arabinanases, exo-α-D-Ara*f*-ase, and exo-β-D-Ara*f*-ase by determining their crystal structures.

## Results

### Identification and characterization of endo-D-arabinanases from *Mi. arabinogalactanolyticum*

*My. smegmatis* AG (MsAG) has been previously reported to induce secretion of endo-D-arabinanase by *Mi. arabinogalactanolyticum* (EndoMA)[19]. The endo-D-arabinanase-active fractions were separated and purified using two-column chromatography until a single band representing a native EndoMA (nEndoMA) protein remained (Figs. 1b and 2). *My. smegmatis* LAM (MsLAM) was degraded into lipomannan (LM), and its enzymatic activity was evaluated using silver periodic acid-Schiff (PAS)-stained SDS-PAGE. High-performance anion-exchange chromatography with pulsed amperometric detection (HPAEC-PAD) analysis indicated that nEndoMA exhibited time-dependent degradation of MsLAM and released soluble arabinooligosaccharides (Supplementary Fig. 1). nEndoMA was also active towards *My. tuberculosis* LAM (MtLAM) (Fig. 1a, c, d), which has a capping structure different from that of MsLAM[6] (Fig. 2a). Our attempts to establish the internal amino acid sequence of the isolated protein using MS/MS analysis were unsuccessful. Therefore, the draft genome of *Mi. arabinogalactanolyticum* was utilized to identify candidate genes for endo-D-arabinanase (Supplementary Table 1) based on protein size and isoelectric point (pI) (Supplementary Fig. 2). From 3458 coding sequences, we identified three putative extracellular proteins with 300–400 amino acids and theoretical pIs of 4.3–4.6. We focused on MIAR_33220, which has a domain of unknown function[27], DUF4185, because two candidates were probably non-enzymatic proteins (OmpA and PBP2). Homologous genes of MIAR_33220 were conserved in the genomes of *My. smegmatis* and *Cellulomonas* sp., from which endogenous endo-D-arabinanase activity has been reported[22,23] (discussed below). Furthermore, DUF4185 proteins are also present in the order *Corynebacteriales* (i.e., *Corynebacteria*,

*Rhodococcus, Gordonia*, and *Nocardia*), which have cell wall D-arabinan (https://pfam.xfam.org/family/DUF4185). Three additional DUF4185 genes, a putative ABC transporter, two putative GH genes (GH172 and GH116), and putative downstream metabolic enzymes were identified close to the MIAR_33220 gene (Fig. 1e). MIAR_33200 encodes a putative enzyme belonging to GH172, for which we previously found exo-α-D-Ara*f*-ase activity[17]. Very recently, the CAZy database created a new GH183 family for DUF4185 enzymes, according to a report for D-arabinan-degrading enzymes from *D. gadei*[25]. The substrate-binding protein (SBP) component gene of the putative ABC transporter (MIAR_33310) belongs to SBP_bac_8 family (PF13416) in the InterPro (Pfam) database. SBP_bac_8 contains maltodextrin binding protein and many other SBPs of sugar transporters. In the Protein Data Bank, MIAR_33310 showed highest sequence similarity to a sugar transporter ATU4361 (PDB ID: 4QRZ, sequence identity = 26%) and cyclic alpha-maltosyl-1,6-maltose binding protein (PDB ID: 7BVT, sequence identity = 26%)[28], suggesting that the putative ABC transporter (MIAR_33290–MIAR_33310) functions as a sugar importer.

Gene expression of the putative polysaccharide utilization locus (PUL) was investigated (Supplementary Fig. 3). *Mi. arabinogalactanolyticum* was cultured in an induction medium containing either Glc, D-Ara, or mycobacterial cell wall extracts (MCE) as the sole carbon source, and gene expression of the cells harvested at the middle exponential growth phase was measured by quantitative real-time PCR (qRT-PCR). The four DUF4185 (GH183) genes (MIAR_32220, MIAR_33230, MIAR_33270, and MIAR_33320) and a GH116 gene (MIAR_33170) were strongly induced by D-Ara and MCE (>10-fold increase compared with Glc condition). The gene for an SBP of the putative ABC transporter (MIAR_33310) was also significantly induced by D-Ara and MCE. Induction of the GH172 gene (MIAR_33200) was relatively mild, but it increased 3.5- and 6.0-fold by D-Ara and MCE, respectively, compared with the Glc condition. This result suggests that the gene cluster containing putative enzymes and an ABC transporter constitute a PUL for LAM and AG degradation. The D-arabinan-degradation PUL of *Mi. arabinogalactanolyticum* seems to be typical for gram-positive bacteria, which is different from that for gram-negative bacteria such as *Bacteroides* species[29].

We heterologously expressed four DUF4185-containing genes that share only moderate amino acid sequence similarity (22–35%) (Supplementary Fig. 4). Although two proteins (MIAR_33220 and MIAR_33320) did not express as soluble proteins in *Escherichia coli*, pure recombinant proteins MIAR_33230 (EndoMA1) and MIAR_33270 (EndoMA2) were obtained (Supplementary Fig. 5a, b). Similar to that of the native enzyme, EndoMA1 and EndoMA2 showed degradation activity towards MtLAM and MsAG (Figs. 1c, d and 3). HPAEC-PAD analysis revealed that EndoMA1 released shorter fragments compared to EndoMA2 and nEndoMA. EndoMA1 and EndoMA2 showed optimum activity at neutral pH for the MtLAM substrate (Supplementary Fig. 5c, d). EndoMA1 had a higher optimal temperature compared with EndoMA2. We synthesized four compounds of D-arabinan substructures with an acetonide tag to investigate the substrate specificities of EndoMA1 and EndoMA2 (Fig. 4 and Supplementary Methods). A22βT is the largest docosasaccharide (22-mer) that has Arabinan motif with Branches, β-linkage, and an acetonide Tag[12]. Linear nonasaccharides, branched octasaccharides, and branched pentasaccharides were designated as A9LT, A8BT, and A5BT, respectively. The degradation patterns of the synthetic substrates by EndoMA1 and EndoMA2 were analyzed using HPAEC-PAD and high-performance liquid chromatography (HPLC) after fluorescent reducing end-labeling (Supplementary Fig. 6). EndoMA1 produced smaller fragments than EndoMA2. For example, EndoMA1 could release D-arabinose by the further hydrolyzation of A8Bβ from A22βT, A4L from A9LT, and A5B from A8BT. In contrast, these larger products persisted in the EndoMA2 digests, and little D-arabinose was released. These results suggested that EndoMA1 has a loose recognition for subsite +1. The

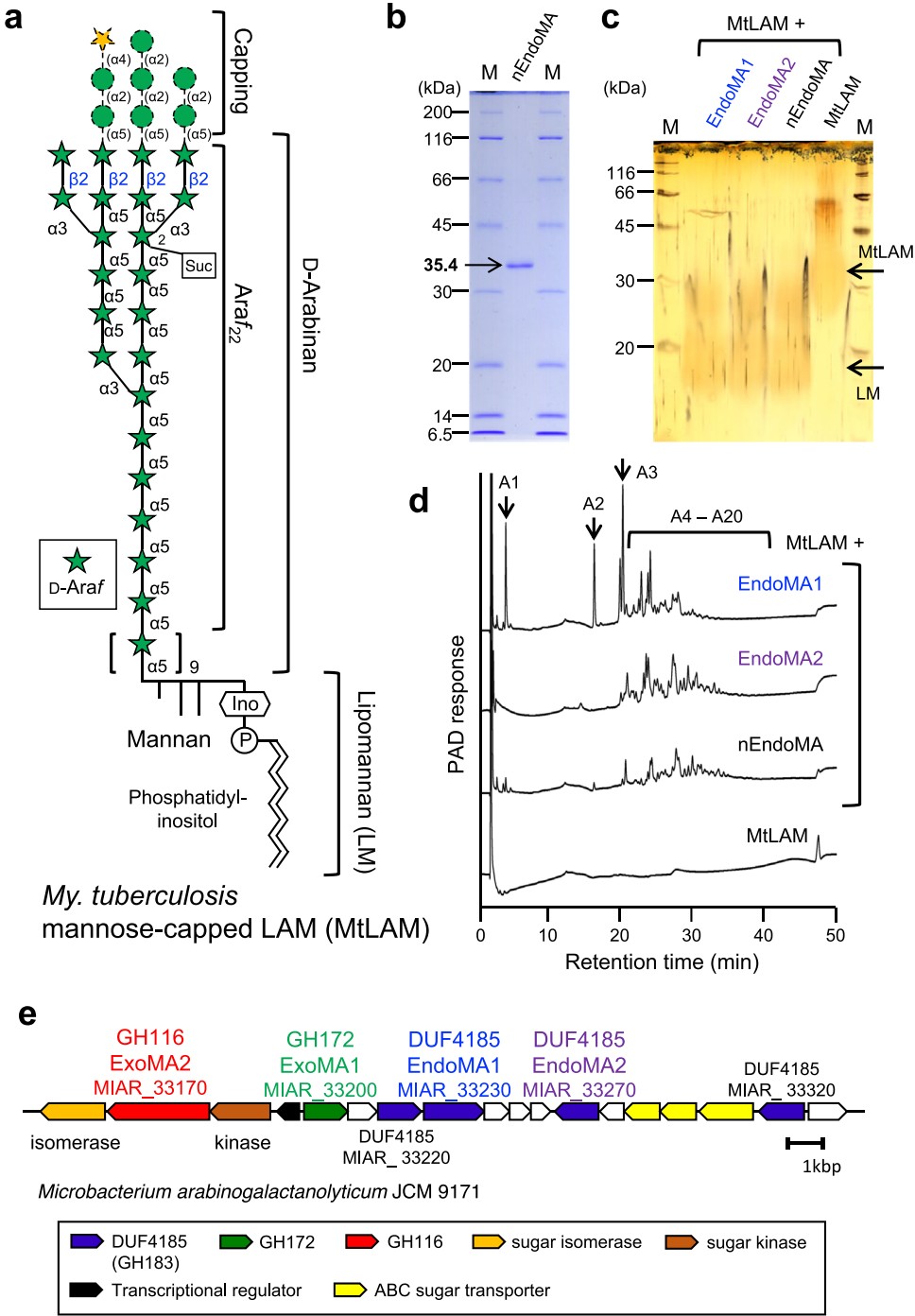

**Fig. 1 | Endo-ᴅ-arabinanases identified from *Mi. arabinogalactanolyticum*.**
**a** A structural model of LAM from *My. tuberculosis* (MtLAM). The ᴅ-arabinan domain consists mainly of α-links, but there are β-links on the non-reducing terminal side. Mannose residues cap the β-linked ᴅ-Ara*f* in MtLAM. LM, lipomannan domain. **b** SDS-PAGE of purified native endo-ᴅ-arabinanase from *Mi. arabinogalactanolyticum* (nEndoMA) and stained by Coomassie brilliant blue. M, protein molecular weight marker. **c**, **d** Degradation of MtLAM analyzed using SDS-PAGE with silver-PAS staining (**c**) and HPAEC-PAD (**d**). MtLAM was incubated with each enzyme in 50 mM sodium phosphate (pH 6.0) at 37 °C overnight. A1-A20 in (**d**) indicates substrate-released arabinooligosaccharides. Numbers represent the degree of polymerization. **e** The ᴅ-arabinan degradation PUL of *Mi. arabinogalactanolyticum*.

products of A22BβT degradation by EndoMA1 were also analyzed using electrospray ionization time-of-flight mass spectrometry (ESI-TOF MS) (Supplementary Fig. 7). MS analysis detected trace levels of product fragments that were not detected by HPAEC-PAD and HPLC analysis. EndoMA1 could attack the central linear region of A22βT, but not the non-reducing end heptasaccharide containing the β-linkage (Fig. 4a). The α-methyl glycoside was synthesized from A5BT with EndoMA1 by transglycosylation activity in the presence of methanol

(Fig. 5a, Supplementary Fig. 8, and Supplementary Table 2), demonstrating that DUF4185 (GH183) endo-ᴅ-arabinanases are anomer-retaining GHs.

## Identification and characterization of exo-ᴅ-arabinofuranosidases

After the discovery of arabinooligosaccharide-releasing endo-ᴅ-arabinanases, we speculated that the putative ᴅ-arabinan degradation PUL

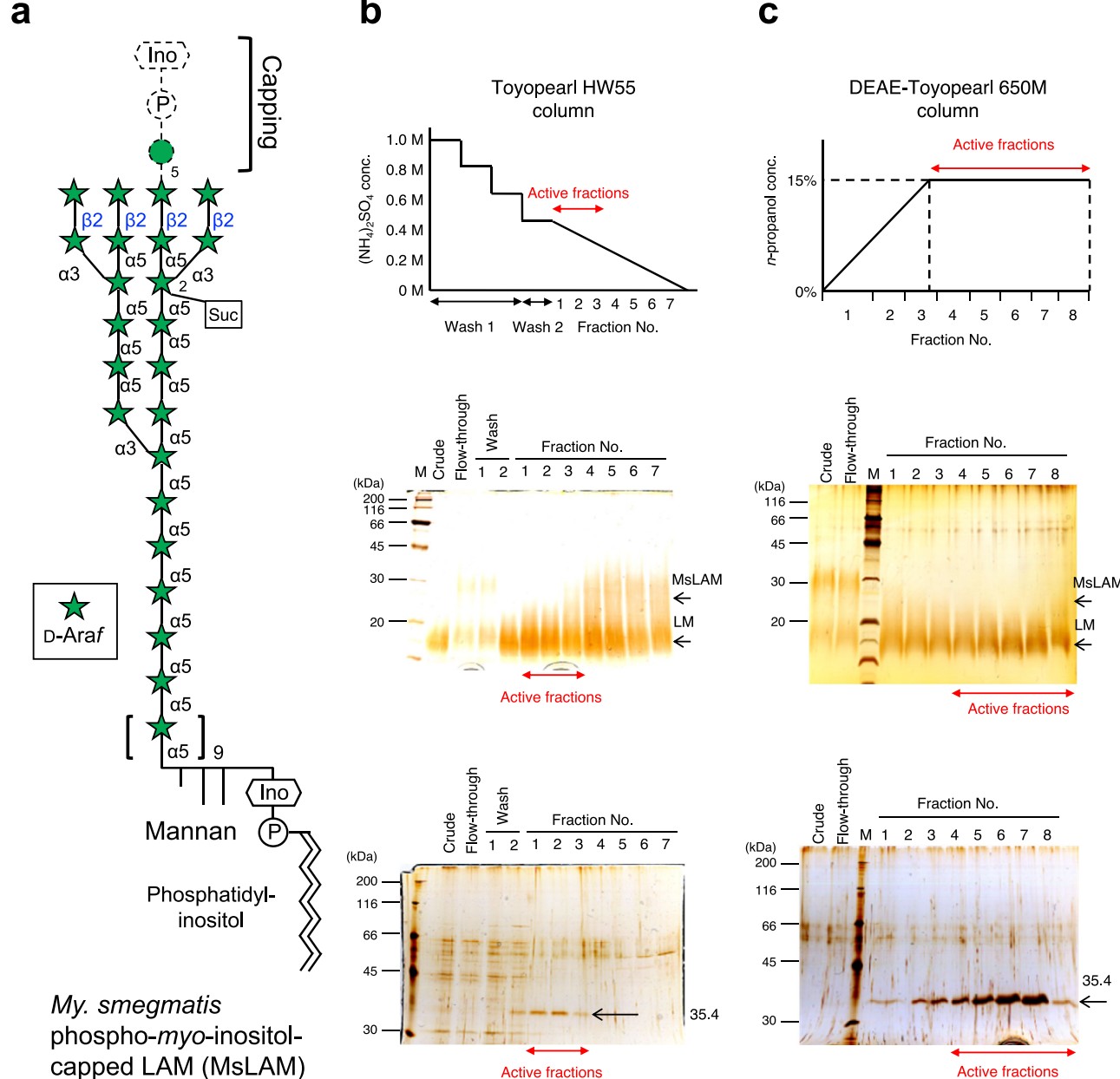

**Fig. 2 | Purification steps of native endo-ᴅ-arabinanase. a** A structural model of LAM from *My. smegmatis* (MsLAM) which was used as the substrate for enzyme purification. **b**, **c** Chromatographies using Toyopearl HW55 (**b**) and DEAE-Toyopearl 650 (**c**) columns. Upper panel: Protein elution conditions. Middle: silver-PAS staining of the reaction product following SDS-PAGE (activity assay). Lower panel: silver staining of protein samples following SDS-PAGE. The crude, flow-through, and eluted fractions are shown. M: protein molecular weight marker.

contains exo-ᴅ-arabinofuranosidases for both the α- and β-bonds (Fig. 1e and Supplementary Fig. 4a). Therefore, we investigated two putative GH genes in the PUL. Recombinant MIAR_33200 (ExoMA1, GH172) and MIAR_33170 (ExoMA2, GH116) proteins were expressed in *E. coli* and purified (Supplementary Fig. 9a, b). Using *p*NP-α-ᴅ-Ara*f* and *p*NP-β-ᴅ-Ara*f* as substrates, ExoMA1 and ExoMA2 were identified as exo-α-ᴅ-Ara*f*-ase and exo-β-ᴅ-Ara*f*-ase, respectively (Fig. 5b, d). ExoMA1 had an optimal pH of 5.5 (Supplementary Fig. 9c) and was specific for the α-ᴅ-Ara*f* bond (Supplementary Fig. 10a). The $K_m$ and $k_{cat}$ values of ExoMA1 at 37 °C were 2.55 mM and 44.4 s⁻¹, respectively, (Supplementary Fig. 10b), which were comparable to those of GH172 αFFase1 ($K_m = 2.71$ mM and $k_{cat} = 127.5$ s⁻¹)[17]. Compared with αFFase1, ExoMA1 showed higher specificity for the α-ᴅ-Ara*f* bond and weaker α-ᴅ-Fru*f*-ase activity (Fig. 5c). As expected from the anomer-retaining GH172 protein, ExoMA1 exhibited transglycosylation activity (Fig. 5e).

ExoMA2 was specific for the β-ᴅ-Ara*f* bond and showed no activity against *p*NP-β-ᴅ-Glc*p* or other *p*NP-substrates (Supplementary Fig. 11a). ExoMA2 had an optimum pH of 6.0 for *p*NP-β-ᴅ-Ara*f* (Supplementary Fig. 9d), with $K_m$ and $k_{cat}$ values of 2.73 mM and 12.0 s⁻¹, respectively, at 37 °C (Supplementary Fig. 11b). ExoMA2 catalyzed transglycosylation of *p*NP-β-ᴅ-Ara*f* with methanol producing Me-β-ᴅ-Ara*f* (Fig. 5f and Supplementary Fig. 12a). Additionally, the stereochemistry of glycosidic bond hydrolysis was monitored using ¹H NMR (Supplementary Fig. 12b). Under equilibrium conditions, *p*NP-β-ᴅ-Ara*f* was hydrolyzed to ᴅ-arabinose as a mixture of α/β-furanoses and pyranoses via mutarotation. After 1 min of hydrolysis, a signal of H-1 of β-ᴅ-Ara*f* appeared as the initial furanose without the appearance of the H-1 signal of α-ᴅ-Ara*f*. In contrast, GH172 αFFase1 (anomer-retaining α-ᴅ-Ara*f*-ase)[17] initially produced α-ᴅ-Ara*f*. This finding indicated that ExoMA2 is an anomer-retaining GH. We then examined the synergistic

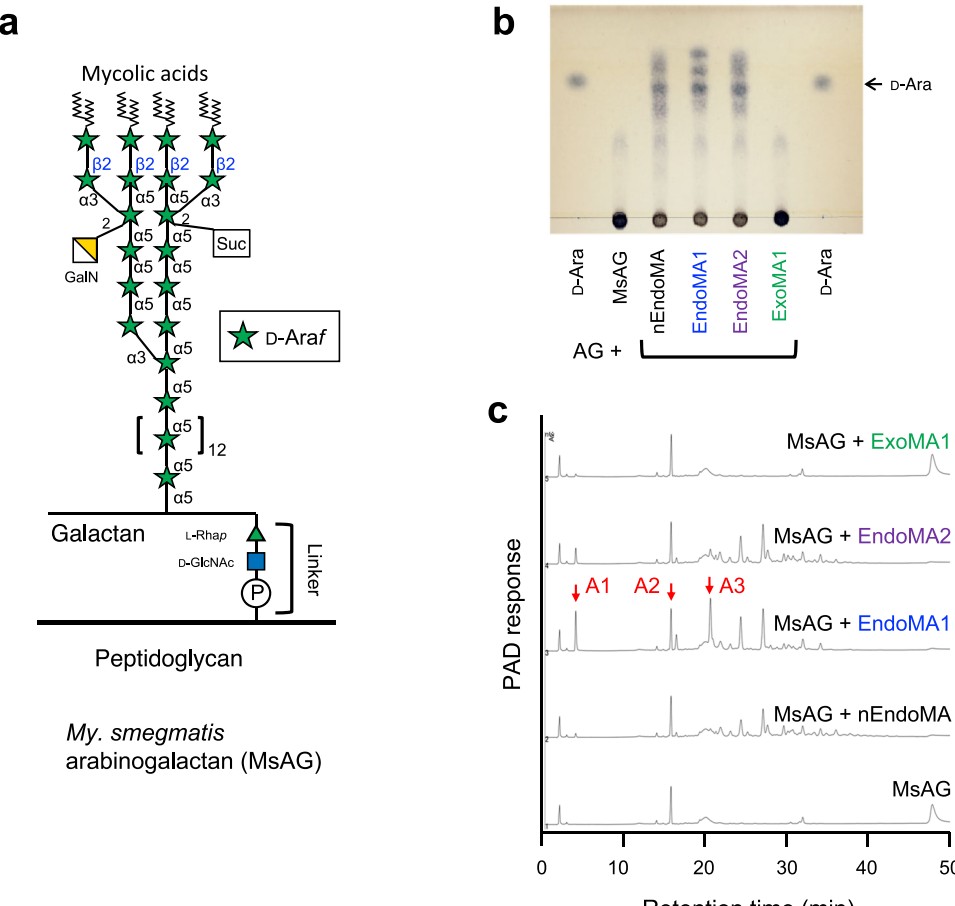

**Fig. 3 | Degradation of mycobacterial arabinogalactan by the native and recombinant ᴅ-arabinan degrading enzymes. a** Structural model of arabinogalactan from *My. smegmatis* (MsAG). **b, c** Analysis of arabinooligosaccharides released from the substrate. **b** TLC analysis. **c** HPAEC-PAD analysis. MsAG (0.50 mg/mL) in 50 mM sodium acetate (pH 6.0) was incubated with each enzyme (1.0 µg/mL) at 37 °C for 20 h. A1–A3 indicate arabinooligosaccharides with DPs 1–3.

action of EndoMA1, ExoMA1, and ExoMA2 on the natural and synthetic ᴅ-arabinan polysaccharides MsLAM and A22BβT (Supplementary Fig. 13). ExoMA1 hydrolyzed the terminal α-ᴅ-Ara*f* structures of arabinooligosaccharides released by EndoMA1 and EndoMA2 but did not act on MsLAM, A22BβT, and A7Bβ. ExoMA2 hydrolyzed the terminal β-ᴅ-Ara*f* structures of MsLAM, A22BβT, and A7Bβ. In summary, ExoMA1 and ExoMA2 synergistically and completely degraded natural and synthetic ᴅ-arabinan substrates into monomers when combined with EndoMA1.

## Crystal and solution structures of EndoMA1

The crystal structures of EndoMA1 were determined at resolutions of 1.60 and 1.80 Å for the apo and ligand complex forms, respectively (Supplementary Table 3). A complex-form crystal was prepared using a catalytic residue mutant (D51N) for the co-crystallization of A9LT. We focused on the complex structure because the apo and complex structures are nearly identical (Cα root mean square deviation (RMSD) < 0.27 Å for all chain pairs). In the catalytic domain, the active site on the front side of a 5-bladed β-propeller fold contained a linear tetrasaccharide with an acetonide tag (A4LT) (Fig. 6a and Supplementary Fig. 14a). Unexpectedly, a linear tetrasaccharide without the tag (A4L) bound to a putative carbohydrate-binding module (CBM) domain that adopted a β-sandwich fold (Supplementary Fig. 14b) was revealed. The EndoMA1 crystal contains four chains (A-D) in the asymmetric unit, and the A4L molecules on the putative CBM domain were only observed at the interface of chains A and C (Supplementary Fig. 14c). The PISA server[30] predicted that the biological assembly of the protein was a dimer of A-B or C-D chains.

Due to elution retardation, size-exclusion chromatography (SEC) using a Superdex column, a dextran-agarose matrix, could not accurately determine the molecular mass of EndoMA1 (Supplementary Fig. 15). Therefore, we performed experiments of multi-angle static light scattering and refractive index combined with SEC (SEC-MALS/RI) and small-angle X-ray scattering combined with SEC (SEC-SAXS) on EndoMA1 to clarify its quaternary structure in solution. The molar masses estimated by SEC-MALS/RI and SEC-SAXS were 105,200 Da and 110,914 Da, respectively (Supplementary Fig. 16a and Supplementary Data 1). The molar mass estimates of monomeric EndoMA1 calculated by the amino acid sequence and measured by SDS-PAGE were 53,678 Da and 52.4 kDa (Supplementary Figs. 5a and 15b), respectively, suggesting that EndoMA1 formed a dimer in solution. The experimental SAXS profile (Fig. 6d, black open circle) and theoretical SAXS profile calculated from the dimeric crystal structure (Fig. 6d, red line) were in good agreement. The $R_g$ obtained from the Guinier analysis and the pair distance distribution function (PDDF) were $31.2 \pm 0.1$ and $31.1 \pm 0.1$ Å, respectively, almost consistent with that calculated from the crystal structure (31.0 Å) (Supplementary Fig. 16b–d). The maximum dimension of the molecule ($D_{max}$) derived from the PDDF was 104 Å, which was also close to that of the crystal structure (98 Å). The normalized Kratky plot in Supplementary Fig. 16e shows a bell-shaped peak with a height of $3\,e^{-1}$ for $QR_g = \sqrt{3}$, along with a shoulder peak around $QR_g = 4$. The profile in the high-Q region is also asymptotic to zero with gradual oscillation, suggesting that EndoMA1 is a globular protein consisting of two domains[31]. The outer shape of the bead model for the solution structure obtained by ab initio modeling is displayed in Fig. 6e and superimposed

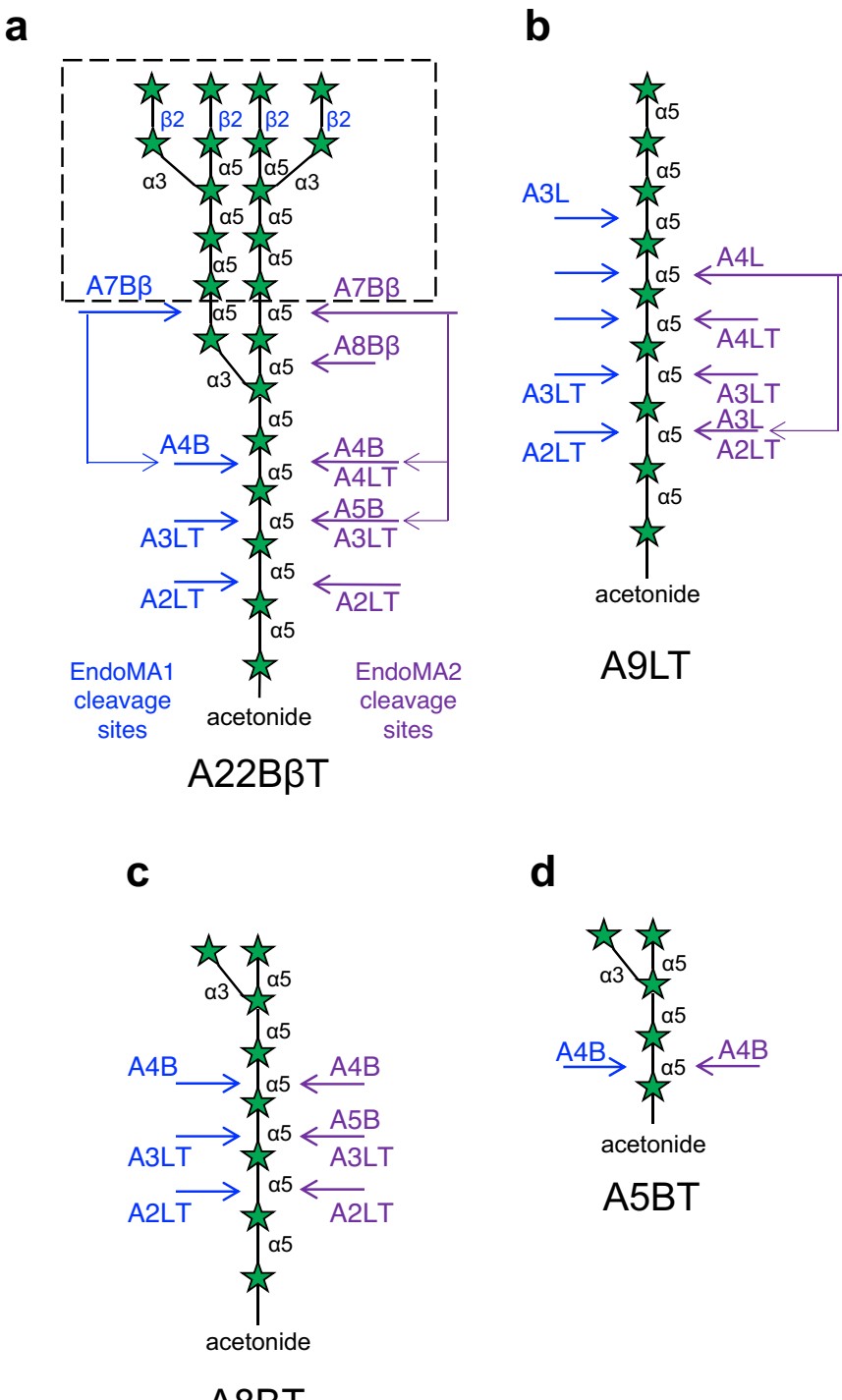

**Fig. 4 | Synthetic oligo-ᴅ-arabinofuranoside substrates used in this study.**
**a** Branched docosasaccharide (A22BβT). ESI-TOF MS analysis revealed that the branched heptasaccharide at the non-reducing end (A7Bβ) is resistant to EndoMA1 (dotted box, Supplementary Fig. 7). **b** Linear nonasaccharide (A9LT). **c** Branched octasaccharide (A8BT). **d** Branched pentasaccharide (A5BT). These oligosaccharides are modified with an acetonide tag. The letters A, B, L, β, and T in the substrate names represent arabinose, branched, linear, β-Ara*f* bond, and acetonide tag, respectively. The main cleavage sites for EndoMA1 and EndoMA2 analyzed by HPAEC-PAD are indicated on the left and right, respectively. Names of the degradation products with and without an acetonide tag are indicated below and above the arrows, respectively.

on the high-resolution crystal structure. These two structural models and their respective SAXS profiles shown in Fig. 6d and Supplementary Fig. 16f were almost identical. These results revealed that EndoMA1 forms a dimer in solution.

In the crystal structure, the active site cleft of the catalytic domain of EndoMA1 adopts the shape of a typical endo-type enzyme (Fig. 6a,

right). The bound A4LT molecule in the complex crystal form spans subsites −3 to +1, and the arabinofuranose ring at subsite −1 adopts a $^3E$ conformation to hold the α-anomeric scissile bond in the pseudoaxial position (Fig. 6b and Supplementary Fig. 14a). The side-chain oxygen of Asp33 is located 2.8 Å from the anomeric C1 atom, making it suitable for in-line nucleophilic attack. Asn51(Asp) forms a hydrogen bond with

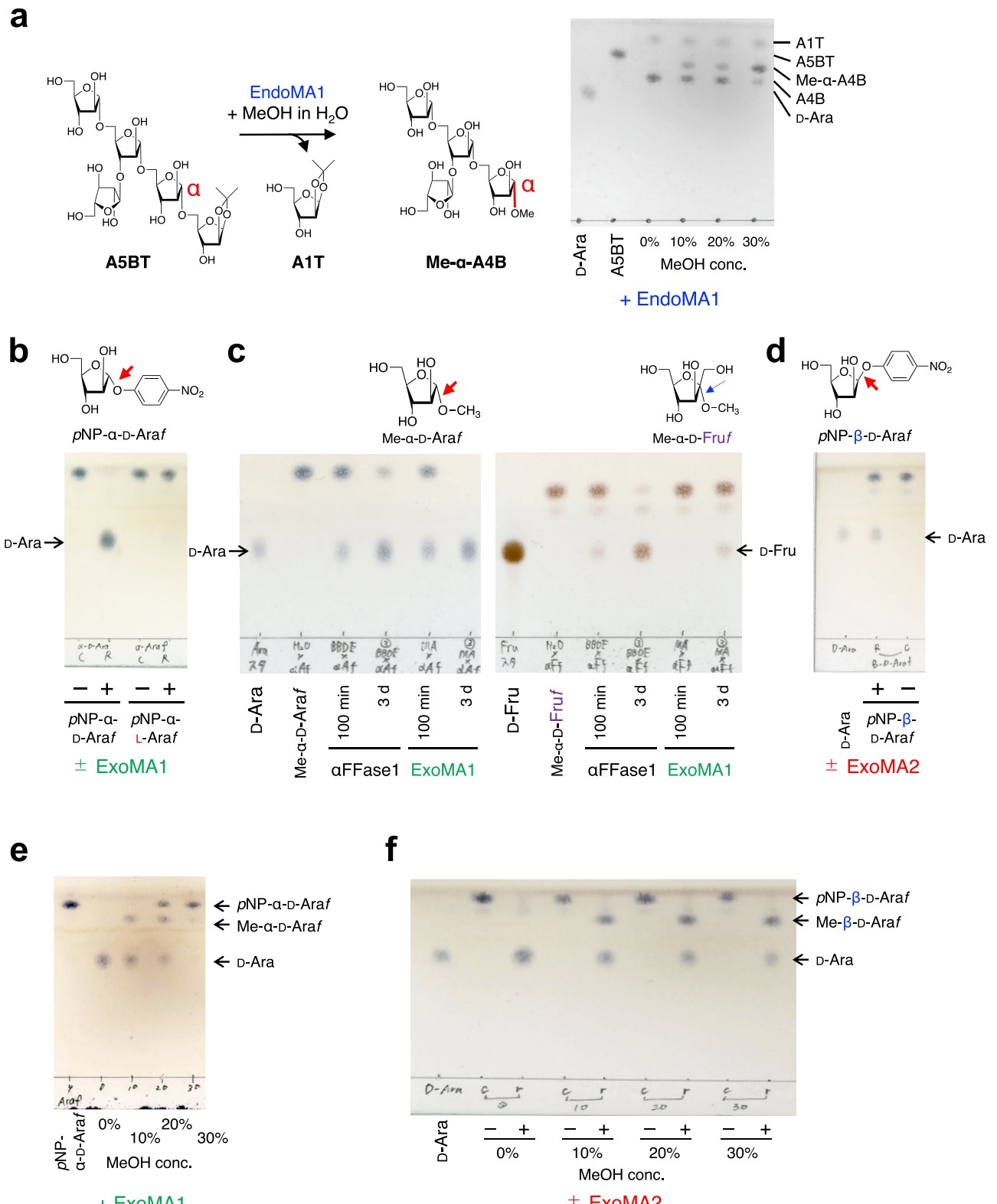

**Fig. 5 | TLC analysis of transglycosylation and substrate specificity of the D-arabinan degrading enzymes. a** Transglycosylation activity of EndoMA1 in the presence of methanol. **b** ExoMA1 activity towards *p*NP-α-D-Ara*f* and *p*NP-α-L-Ara*f*. **c** Activity of GH172 enzymes (ExoMA1 and αFFase1) towards Me-α-D-Ara*f* and Me-α-D-Fru*f*. **d** Activity of ExoMA2 towards *p*NP-β-D-Ara*f*. **e, f** Transglycosylation activities of ExoMA1 (**e**) and ExoMA2 (**f**) in the presence of methanol. Detailed assay conditions are described in the "Methods" section.

the glycosidic bond oxygen between subsites −1 and +1, suggesting that it is the catalytic acid/base residue. Pro141, Arg198, and Leu292 form hydrogen bonds with A4LT, while Tyr293 and Trp138 form stacking interactions above and below the cleft. Glu243 is the third Asp/Glu residue in the active site cleft and supports the side chain position of Arg198, which plays a pivotal role in holding sugars at subsites −3 and −2 (Supplementary Fig. 17a). The activity of the D33N, D51N, E243A, and E243Q mutants toward A22BβT was nearly abolished

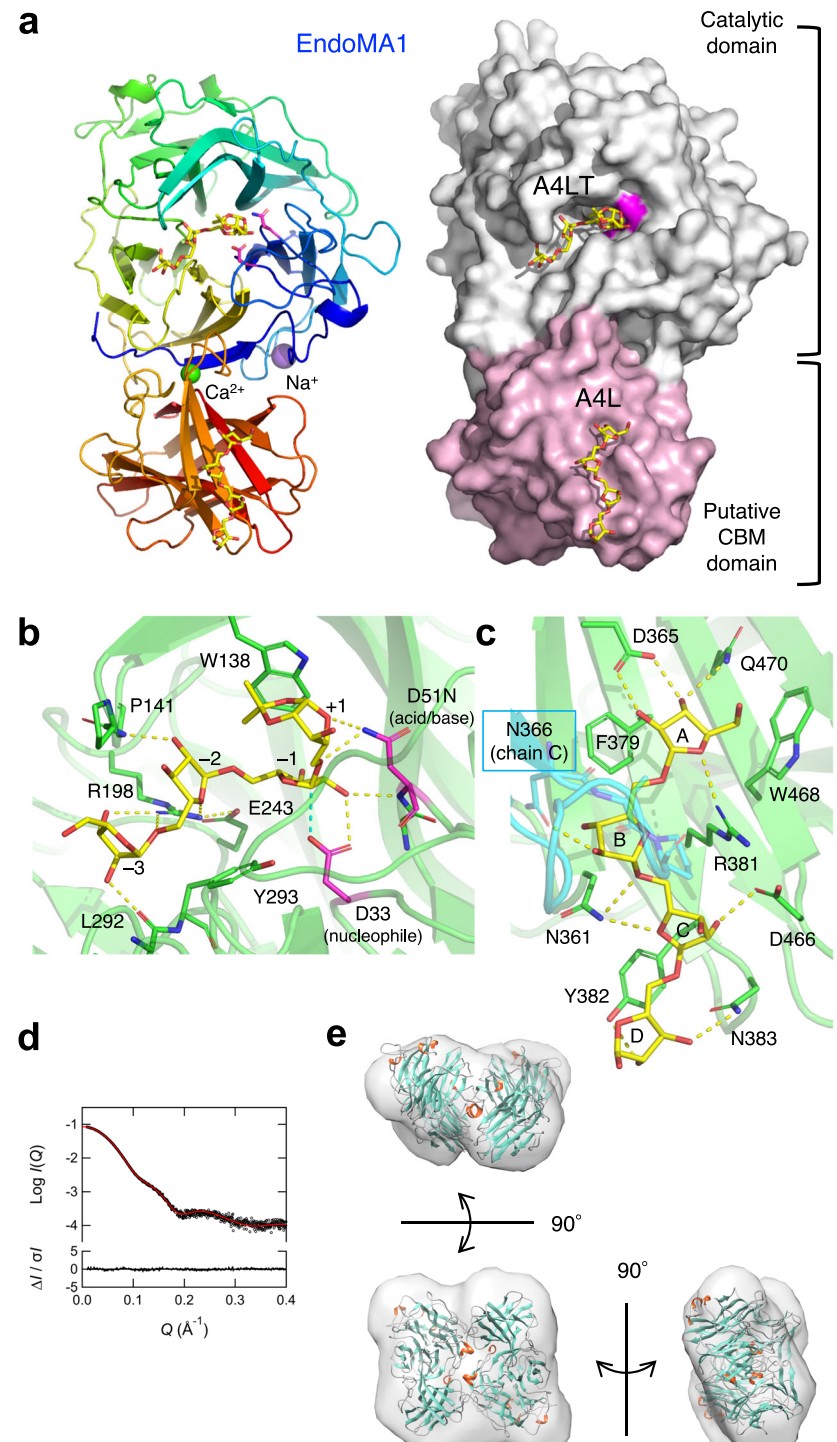

**Fig. 6 | Three-dimensional structures of EndoMA1. a–c** The crystal structure in complex with linear oligo-D-arabinofuranosides. In EndoMA1 co-crystallized with A9LT, A4LT (tetrasaccharide with acetonide tag) was observed in the active site, whereas A4L (tetrasaccharide without acetonide tag) was observed in the putative CBM domain. **a** Overall structure. A ribbon model (left) and the molecular surface (right) are shown. The catalytic residues are represented in magenta. **b** The active site with bound A4LT. **c** The binding site of A4L within the putative CBM domain. **d, e** SAXS analysis. **d** Superimposition of the theoretical (red) and experimental (black) SAXS profiles. The theoretical profile was calculated from the crystal structure in the apo form. The residuals of both profiles are drawn at the bottom of the graph. **e** The outer shape of the bead model for the solution structure is superimposed on the dimeric crystal structure.

(Supplementary Fig. 17b), confirming that all three Asp/Glu residues are catalytically essential.

The A4L molecule is bound to the shallow surface of the putative CBM domain (Fig. 6a, right). Among the four subsites identified in the crystal structure (A-D), subsite A has the strongest interactions (Fig. 6c). The arabinose unit at subsite A is sandwiched between the aromatic side chains of Phe379 and Trp468, and hydrogen bonds from Asp365, Arg381, and Gln470 are formed. Asn361, Asn383, and Asp466 form hydrogen bonds with the sugars in subsites B, C, and D, and Tyr382 forms a stacking platform for subsites C and D. Asn366 from the neighboring molecule (chain C) is involved in the interaction with A4L, but this seems to be a crystal packing artifact (Supplementary Fig. 14c).

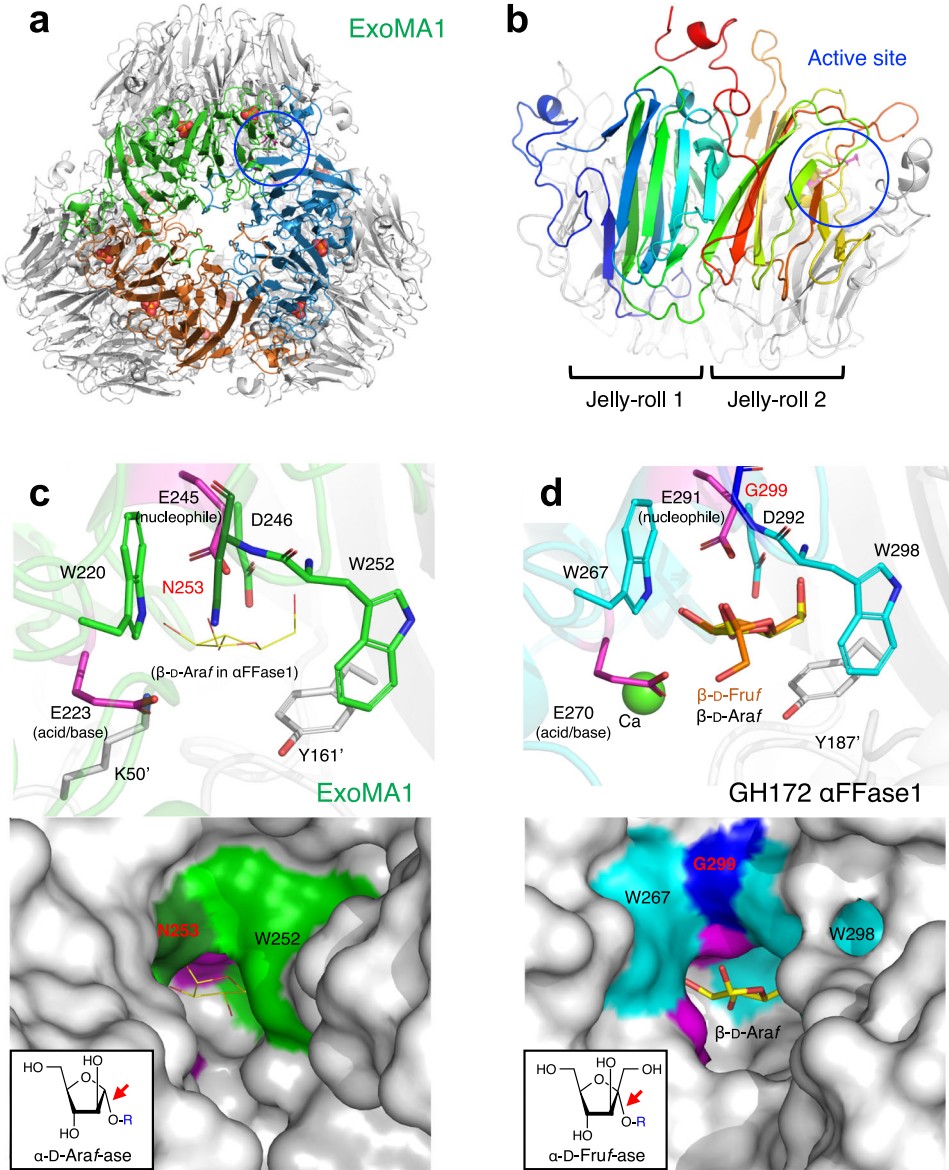

**Fig. 7 | Crystal structure of ExoMA1. a** Overall structure of the dodecamer in the asymmetric unit. **b** Monomer structure. One protomer within the trimer unit is depicted in rainbow color. **c, d** The active sites of ExoMA1 (**c**) and GH172 αFFase1 (**d**). A ribbon model (upper panel) and the molecular surface (lower panel) are shown. In (**c**), the β-D-Araf molecule bound to αFFase1 is superimposed on the ExoMA1 structure and depicted as thin yellow sticks. In (**d**), β-D-Fruf and β-D-Araf molecules bound to αFFase1 are superimposed and represented by yellow and orange sticks, respectively. Residues from a neighboring protomer are labeled with prime.

A Dali structural search showed that the catalytic domain of EndoMA1 was similar to the hypothetical protein BACOVA_04882, *N*-acetylgalactosamine deacetylase, and GH43 α-L-Araf-ases (Supplementary Table 4). A structural comparison with a GH43 α-L-Araf-ase from *Cellvibrio japonicus*[32] indicated that the nucleophile of EndoMA1 (Asp33) is located in the same topological position as the catalytic base of the inverting GH43 enzyme (Asp41), whereas the acid/base (Asp51) is differently located from the catalytic acid (Glu215) of GH43 (Supplementary Fig. 18a, b). The putative CBM domain is structurally similar to CBM4-2 in GH10 xylanase[33] and CBM61 in GH31 α-1,6-glucosyltransferase[34] (Supplementary Table 4). The D-arabinan binding site of EndoMA1 is not on the concave side of the β-sandwich fold, where the xylan binding site of CBM4-2 is located (Supplementary Fig. 18c, d). CBM61 has dual binding sites for maltooligosaccharides, one of which (the B61-2 site) corresponds to the putative CBM binding site in EndoMA1. Currently, we lack sufficient evidence to classify this domain as a new CBM. The available data only includes the crystal structure prepared under a high ligand concentration condition, which may have caused potential crystal packing artifacts. Further study is necessary to provide biochemical evidence and to establish this domain as a new CBM family.

## Crystal structures of ExoMA1 and ExoMA2

The crystal structure of ExoMA1 was determined at 2.42 Å resolution in the apo form (Supplementary Table 5). ExoMA1 assembles four trimers to form a dodecamer (Fig. 7a). A SEC experiment suggested that more than heptamer were assembled in solution (Supplementary Fig. 19a). ExoMA1 monomer has a double jelly-roll fold, which was found only in GH172 among over 160 GH families (Fig. 7b)[17,35]. The active site is located at the interface between the two subunits of the basal trimer. One phosphate molecule per monomer was bound to the interface of ExoMA1, which is responsible for dodecamer assembly (Supplementary Fig. 19b). The dodecamer assembly of ExoMA1 was also visible in a negatively stained transmission electron micrograph (Supplementary

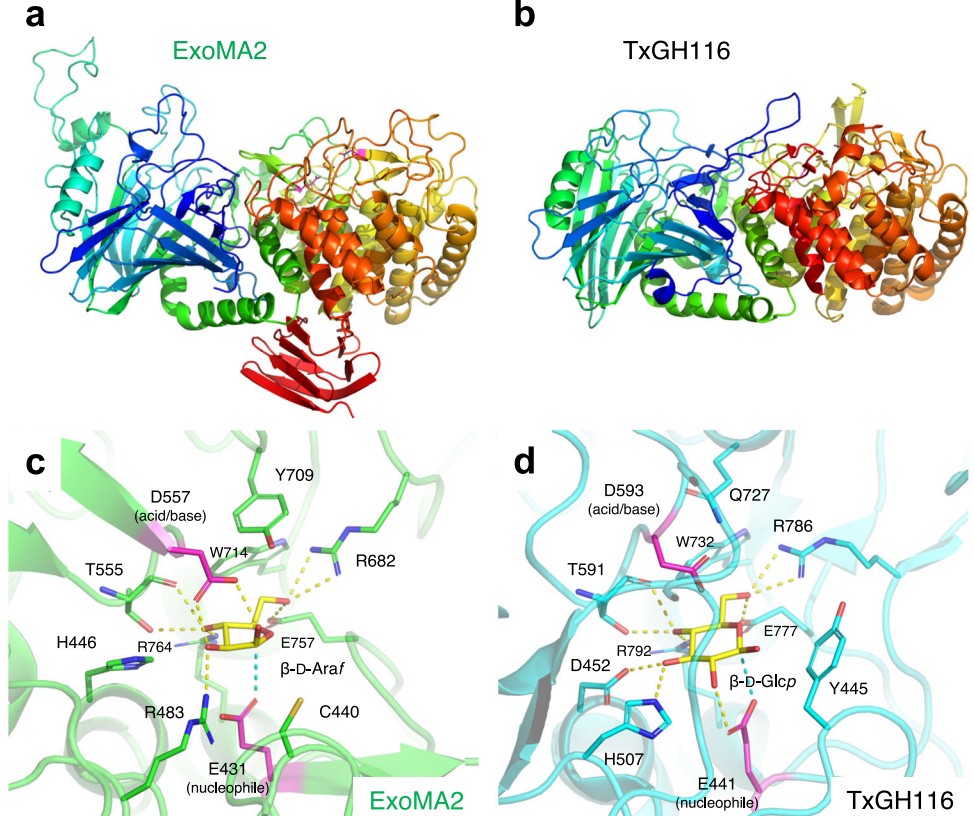

**Fig. 8 | Crystal structures of ExoMA2 and GH116 β-glucosidase. a, b** Overall structures of ExoMA2 (**a**) and GH116 β-glucosidase TxGH116 from *Thermoanaerobacterium xylanolyticum* (**b**). The N-terminal β-sandwich and C-terminal catalytic (α/α)$_6$ barrel domains are represented by a rainbow-colored ribbon model, with the N-terminus in blue and the C-terminus in red. **c, d** The active sites of ExoMA2 (green) complexed with β-D-Ara*f* (**c**) and TxGH116 (cyan) complexed with β-D-Glc*p* (**d**). Magenta and yellow sticks, respectively, represent the catalytic residues and the ligands. Hydrogen bonds are shown as yellow dotted lines, and the distance between the anomeric carbon of the ligand and the nucleophile is represented as a cyan dotted line.

Fig. 19c). Other GH172 members (αFFase1 and a hypothetical protein BACUNI_000161 from *Bacteroides uniformis*) had been shown to form distinct quaternary structures depending on the mode of contact of the basal trimer (Supplementary Fig. 20). The dodecameric structure of ExoMA1 is formed by the interactions of four loops above the double jelly-roll fold, whereas the C-terminal helix is mainly involved in the hexamer (dimer of trimers) interactions in αFFase1 and BACUNI_000161.

Due to the specificity of ExoMA1 and αFFase1 for α-D-Ara*f* and α-D-Fru*f*, respectively (Fig. 5c), their active sites were compared. The anomer-inverted monosaccharides β-D-Ara*f* and β-D-Fru*f* have been bound in the active site of α-D-Fru*f*-ase αFFase1 (Fig. 7d). The β-anomer side of the active site of αFFase1 can accommodate the 1-hydroxymethyl group of Fru*f* because of the presence of the small amino acid Gly299. The corresponding residue in ExoMA1 is Asn253 (Fig. 7c), and its side chain blocks α-D-Fru*f* binding, making the enzyme specific for α-D-Ara*f*.

The crystal structures of ExoMA2 were determined at 1.75 and 1.35 Å resolutions for the complex forms with Tris (buffer molecule) and β-D-Ara*f*, respectively (Supplementary Table 5 and Supplementary Fig. 21a, b). The two structures are almost identical (Cα RMSD < 0.19 Å for all chain pairs), and we will mainly describe the complex structure with β-D-Ara*f*. A dimer is present in the asymmetric unit of the ExoMA2 crystal, consistent with the SEC measurement in solution and the PISA server prediction (Supplementary Fig. 21c, d). ExoMA2 has a two-domain structure consisting of N-terminal β-sandwich and C-terminal (α/α)$_6$ barrel domains, the latter being the catalytic domain (Fig. 8a). The two-domain structure resembles that of GH116 β-glucosidase from *Thermoanaerobacterium xylanolyticum* (TxGH116, Dali Z score = 42.4

and Cα RMSD = 2.5 Å, Fig. 8b)[36], whereas the amino acid sequence identity is very low (21%). Compared with TxGH116, ExoMA2 has an additional long loop and a small β-sandwich domain in the N-terminal and C-terminal domains, respectively. The β-D-Ara*f* molecule is bound to subsite −1 of ExoMA2, and the catalytic nucleophile (Glu431) and acid/base (D557) residues are in the appropriate positions for anomer-retaining hydrolysis (Fig. 8c). All hydroxy groups of β-D-Ara*f* are hydrogen-bonded to the amino acid side chains of the protein, explaining the strict substrate specificity of this enzyme. When compared to GH116 β-glucosidase, the two catalytic residues, and several residues are conserved, but substrate recognition at subsite −1 of ExoMA2 is different. Specifically, residues corresponding to Arg483 and His446 in ExoMA2 are His507 and Asp452 in β-glucosidase TxGH116 (Fig. 8d), which is the key difference in recognition of the O2 hydroxy group of Ara*f* and the O3 hydroxy group of glucopyranose.

## Discussion

In this study, we identified the genes encoding DUF4185 (GH183) endo-D-arabinanases in *Mi. arabinogalactanolyticum*. GH183 family in the CAZy database currently lists 2569 ORFs, all of which have the DUF4185 domain (http://www.cazy.org/GH183.html). DUF4185 (PF13810) in Pfam (InterPro) database currently includes approximately 5,000 members and is distributed among 2225 species of bacteria and fungi. The majority of these 1476 species belong to the phylum *Actinobacteria*, which includes *Microbacterium*, *Mycobacterium*, *Corynebacterium*, *Rhodococcus*, *Nocardia*, and *Cellulomonas*. Not only *Mycobacterium* species but also other bacteria belonging to the order *Corynebacteriales* (i.e., *Corynebacteria*, *Rhodococcus*, *Gordonia*, and *Nocardia*) have D-arabinan in their cell walls[2,37]. Interestingly, *My.*

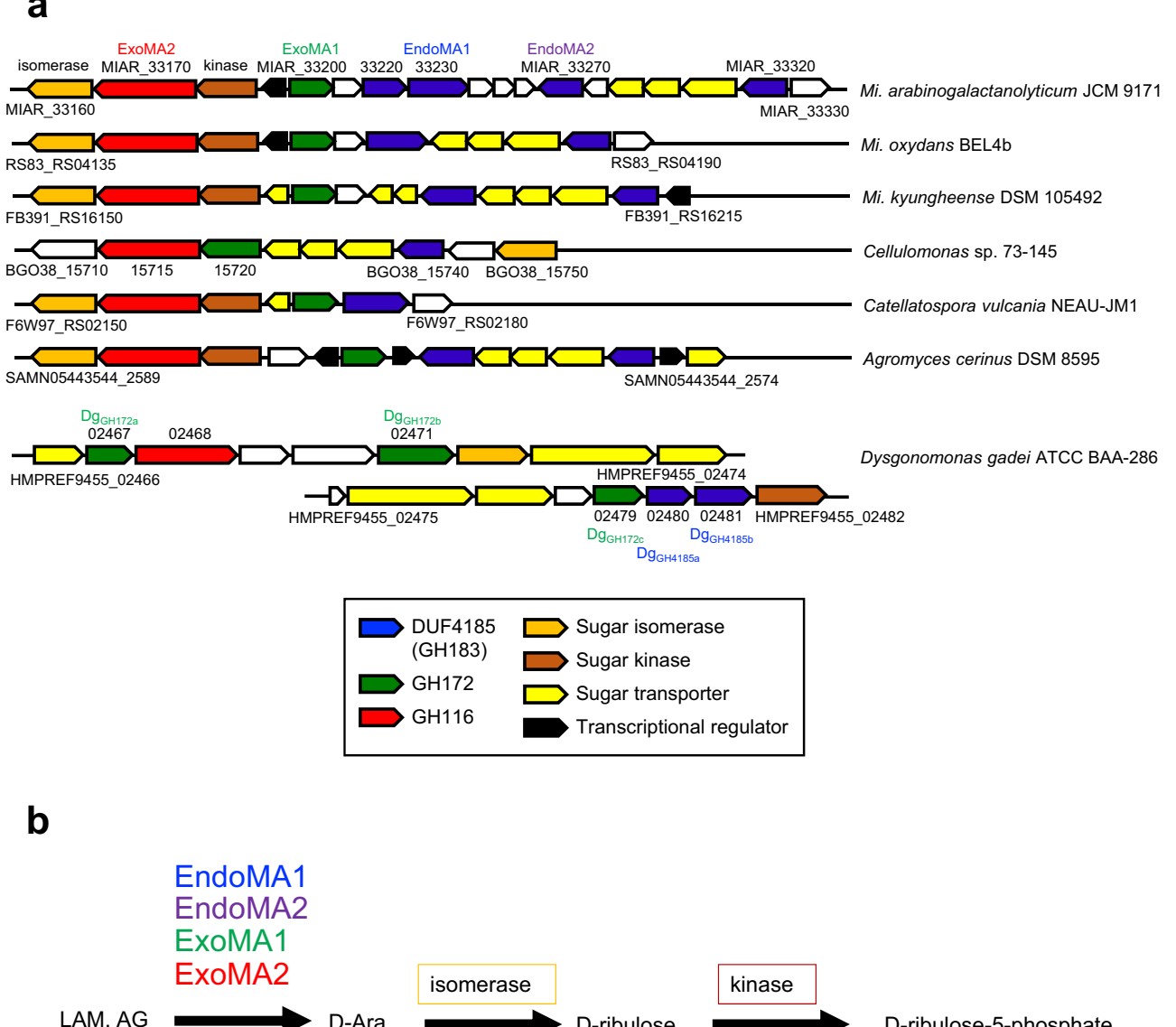

**Fig. 9 | Gene clusters of D-arabinan-degrading enzymes in bacteria. a** D-Arabinan-degrading PUL from *Mi. arabinogalactanolyticum* and related gene clusters in other bacteria. **b** Scheme for degradation of the D-arabinan structures of LAM and AG to D-arabinose, which is then converted to D-ribulose-5-phosphate and metabolized further via the pentose phosphate pathway and Calvin cycle.

tuberculosis H37Rv, and *My. smegmatis* MC² 155 have two and five DUF4185 genes in their genome, respectively[38,39]. Therefore, DUF4815 genes are possibly responsible for producing the endogenous endo-D-arabinanase in *My. smegmatis*[23,40]. In addition to the previously described endo-D-arabinanase[21,22], *Cellulomonas* sp. 73−145 also encodes a gene set of D-arabinan degradative enzymes (DUF4185, GH172, and GH116 candidates) (assembly ASM189818v1, Fig. 9). Putative D-arabinan degradation gene clusters were also present in other bacteria in the phylum *Actinobacteria*. Recently, Al-Jourani et al. reported GH172 exo-α-D-Araf-ases (Dg$_{GH172a}$ and Dg$_{GH172c}$) and DUF4185 (GH183) endo-D-arabinanases (Dg$_{GH4185a}$ and Dg$_{GH4185b}$) from *Dysgonomonas gadei* ATCC BAA-286, which belongs to the phylum *Bacteroidota*[25]. While they did not find an exo-β-D-Araf-ase, the *D. gadei* gene cluster does contain a putative GH116 exo-β-D-Araf-ase gene (HMPREF9455_02468).

A phylogenetic analysis of DUF4185 proteins in the genomes of *Mi. arabinogalactanolyticum*, mycobacteria and other species are shown in Supplementary Fig. 22. EndoMA1 and EndoMA2 are located in different clades, along with homologs from *My. tuberculosis* and *My.*

*smegmatis*, suggesting that members of this clade possess endo-D-arabinanase activity. The C-terminal putative CBM domain in EndoMA1 is conserved only in its closest homologs. Two endo-D-arabinanases from *D. gadei* (DG$_{GH4185a}$ and DG$_{GH4185b}$) are present in the same clade as EndoMA2. MIAR_33320 and MIAR_33220, which are not characterized in this study, were located near the DUF4185 proteins in *Cellulomonas* species. The enzymatic function of MIAR_33220 is elusive because its homolog is not present in the putative D-arabinan degradation PULs of other *Microbacterium* species (Fig. 9 and Supplementary Fig. 22).

Phylogenetic analysis of ExoMA1 and other GH172 enzymes is shown in Supplementary Fig. 23. The distinct substrate preference for α-D-Araf- and α-D-Fruf-bonds suggests that ExoMA1 belongs to a clade distinct from αFFase1. Dg$_{GH172a}$ has α-D-Araf-ase activity[25] and is placed in the same clade as ExoMA1. Although Dg$_{GH172c}$ is highly similar to αFFase1 (α-D-Fruf-ase), there are no reports of α-D-Fruf-ase activity of *D. gadei* enzymes. A phylogenetic analysis of ExoMA2 and related GH116 enzymes is shown in Supplementary Fig. 24 according to the subfamily classification proposed by Ferrara et al.[41]. Most GH116 members are

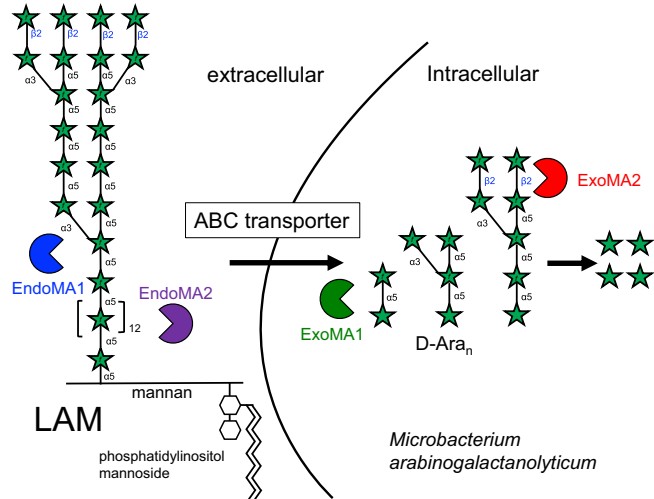

**Fig. 10 | Schematic drawing of the degradation pathway of D-arabinan by Mi. arabinogalactanolyticum.** The two extracellular endo-acting enzymes (EndoMA1 and EndoMA2) depolymerize mycobacterial lipoarabinomannan and arabinoga-lactan, and the ABC transporter presumably imports arabinooligosaccharides. The two intracellular exo-acting enzymes (ExoMA1 and ExoMA2) further degrade ara-binooligosaccharides into D-arabinose monosaccharides, which are metabolized by an isomerase and a kinase.

currently characterized as β-glucosidase (EC 3.2.1.21) and acid β-glu-cosidase/β-glucosylceramidase (EC 3.2.1.45) in bacteria and eukar-yotes, including humans, mice, and *Arabidopsis*, and they are classified in subfamily 1. The archeon *Saccharolobus solfataricus* has two bifunctional enzymes: β-glucosidase/β-*N*-acetylglucosaminidase (EC 3.2.1.52) SSO3039 in subfamily 2 (ref. 41) and β-glucosidase/β-xylosi-dase (EC 3.2.1.37) SSO1353 in subfamily 3 (ref. 42). Ferrara et al. clas-sified bacterial homologs of ExoMA2 in the large subfamily 2, which also included many uncharacterized protein sequences[41]. However, in the phylogenetic tree of the current study (Supplementary Fig. 24), the large clade, including ExoMA2, is separated from the archaeal enzyme clade. Therefore, we designate the large clade including ExoMA2 as a new subfamily 4. Amino acid sequence alignment indicated that all residues that recognize the substrate are conserved in a subfamily 4 member from *Thermobaculum terrenum* ATCC BAA-798 (Tter_0211, GenBank accession number: ACZ41133.1) (Supplementary Fig. 25). Furthermore, all GH116 candidates in the putative D-arabinan-degrad-ing gene cluster belong to the same subfamily 4, suggesting exo-β-D-Ara*f*-ase activity. While the nucleophile residues of subfamilies 1 and 4 did not align with the amino acid sequence alignment due to the very low homology in this region, the catalytic residues of ExoMA2 and TxGH116 are in the same location on the three-dimensional struc-ture (Fig. 8).

We propose a D-arabinan-degrading pathway for *Mi. arabinoga-lactanolyticum* based on its gene cluster and four characterized enzymes (Fig. 10). EndoMA1 and EndoMA2, two new GH family endo-D-arabinanases with signal sequences, depolymerize mycobacterial lipoarabinomannan and arabinogalactan extracellularly. The putative ABC transporter in PUL (Fig. 1e) possibly imports arabinooligo-saccharides. The two exo-enzymes (GH172 ExoMA1 and GH116 ExoMA2) without a signal sequence further degrade arabinooligo-saccharides into monosaccharides intracellularly. Furthermore, the PUL contains a putative isomerase and kinase for D-arabinose meta-bolism (Fig. 9b).

We also determined the crystal structures of the three enzymes. The substrate-complex structure of EndoMA1 clarified the substrate recognition and catalytic mechanism of a novel GH family (GH183) endo-D-arabinanase. ExoMA1 has a unique dodecameric structure compared to the various quaternary structures of the GH172 enzymes[35]. The dodecameric quaternary structure of ExoMA1 is dis-tinct from the hexameric assembly of other GH172 members: αFFase1, BACUNI_000161, and Dg$_{GH172c}$ from *D. gadei* (ref. 25). The discovery of ExoMA2 will lead to the establishment of a new subfamily of exo-β-D-Ara*f*-ase in GH116, and its crystal structure with β-D-Ara*f* revealed substrate recognition distinct from GH116 β-glucosidases.

The presence of DUF4185 (GH183) homologs in mycobacteria suggests the importance of degradative enzymes in the remodeling and recycling of cell wall polysaccharides. The antimycobacterial drug ethambutol, which targets the biosynthesis of D-arabinan, has been widely studied as a potential therapy for tuberculosis and other mycobacterial infections[43]. Data from our genetic, functional, and structural studies on D-arabinan-degrading enzymes will contribute to developing enzymatic tools for the structural and functional analysis of LAM and AG remodeling and recycling in mycobacterial cell walls.

## Methods

### Materials
MtLAM from *My. tuberculosis* Aoyama-B was purchased from Nacalai Tesque, Inc. (Kyoto, Japan). MsLAM from *My. smegmatis* ATCC 700084 was prepared as previously described[44]. AG from *My. smegmatis* ATCC 700084 was prepared using the methods outlined in the previous study[20]. A22BβT was synthesized as reported previously[12]. The synth-esis and schemes of other oligosaccharides and NMR spectra are described in the Supplementary Methods and Supplementary Data 2.

### Purification of native endo-D-arabinanase from *Mi. arabinogalactanolyticum*
Crude endo-D-arabinanase was produced from *Mi. arabinogalactano-lyticum* JCM9171 by inducing with mycobacterial cell wall extracts, as described previously[19]. Culture supernatant was made up to 1.0 M ammonium sulfate and loaded to a Toyopearl HW55 column (12 × 120 mm; Tosoh Corp., Tokyo, Japan) which equilibrated with 1.0 M ammonium sulfate in 50 mM acetate buffer (pH 6.0) as described previously[45]. Step and linear gradients of 1.0, 0.8, 0.6, 0.4 M, 0.4-0 M ammonium sulfate in 50 mM acetate buffer (pH 6.0) were used for elution. Each fraction (43 µl) was mixed with 7 µl MsLAM (4 mg/mL) and 50 µL McIlvain buffer (pH 5.0) and incubated at 37 °C for 16 h. Degradation of LAM to LM was analyzed by SDS-PAGE stained with silver-PAS, as described below. The active fractions were combined at the concentration of 0.4–0.2 M ammonium sulfate and then applied to a DEAE Toyopearl 650 M column (18 × 160 mm; Tosoh Corp.). The column was washed with 1.0 M NaCl and equilibrated with 50 mM acetate buffer (pH 6.0). The protein was eluted with a linear gradient of 0–15% *n*-propanol in 50 mM acetate buffer (pH 6.0), followed by 15% *n*-propanol in 50 mM acetate buffer (pH 6.0). The active fractions were combined and lyophilized. The molecular mass and the isoelectric point of the purified protein were analyzed using SDS-PAGE and two-dimensional electrophoresis using Auto 2D BM-100 (SHARP Corp., Osaka, Japan).

### Draft genome sequencing
*Mi. arabinogalactanolyticum* JCM 9171 was grown under aerobic con-ditions in NBRC 802 medium, and genomic DNA was extracted using the FastPure DNA kit (Takara Bio Inc., Shiga, Japan). The genomic library was prepared using the TruSeq DNA Sample Prep Kit and sequenced using the Illumina HiSeq2000 platform at Hokkaido Sys-tem Science Co., Ltd. (Hokkaido, Japan). Thirteen contigs were assembled using Velvet (version 1.2.08).

### Quantitative real-time PCR
*Mi. arabinogalactanolyticum* JCM 9171 was cultured at 30 °C under aerobic conditions on endo-D-arabinanase induction medium pre-pared with 0.2% casamino acid, 0.005% K$_2$HPO$_4$, and 0.005% MgSO$_4$

$7H_2O$ containing either 0.02% mycobacterial cell wall extracts (MCE), D-arabinose, or glucose. Precipitates of 2 mL culture solutions were harvested at the middle exponential growth phase and stored at −80 °C. The frozen cells were disrupted using a Multi-Beads Shocker MB2200 (Yasui Kikai, Osaka, Japan) operated at 2,000 rpm for 10 s, and were homogenized again using a Multi-Beads Shocker (2,000 rpm, 10 s) after the addition of 1 mL RNAiso Plus (Takara Bio Inc.). The upper phase of the chloroform-extracted samples was purified using an SV Total RNA Isolation System (Promega, Madison, WI). After the removal of DNA contamination by Deoxyribonuclease (RT Grade) for Heat Stop (Nippon Gene, Tokyo, Japan), the total RNA was reverse-transcribed with ReverTra Ace qPCR RT Master Mix (TOYOBO, Osaka, Japan) to synthesize cDNA. Quantitative real-time PCR (qRT-PCR) was performed with Luna Universal qPCR Master Mix (New England BioLabs, MA, USA) using StepOnePlus and StepOne Software version 2.3 (Applied Biosystems, Foster City, CA, USA). Sequences of the specific primer sets used in this PCR were listed in Supplementary Data 3. The relative expression levels of the genes were calculated by ΔΔCt method normalized by RNA polymerase β subunit (rpoB; MIAR_19880).

### Expression and purification of recombinant enzymes

Genes of MIAR_33170, MIAR_33200, MIAR_33220, MIAR_33230, MIAR_33270, and MIAR_33320 were amplified from genomic DNA using PrimeSTAR HS DNA Polymerase (Takara Bio Inc.). The primers were designed to amplify the entire gene without a putative signal sequence. Primers used for gene cloning are listed in Supplementary Data 3. The amplicons were cloned into the pET-23d vector (Novagen, Madison, WI, USA) using an In-Fusion HD cloning kit (Clontech Laboratories Inc., Palo Alto, CA, USA), and a C-terminal His-tag was attached. A SKIK peptide tag was fused to the N-terminus of ExoMA2 to facilitate its expression[46]. E. coli BL21 (DE3) harboring the plasmids were then grown at 25–37 °C using the Overnight Express Autoinduction System (Novagen). After centrifugation, the pellets from the cell culture were resuspended in BugBuster protein extraction reagent (Novagen). The recombinant proteins used for biochemical assays were purified by immobilized metal affinity chromatography (IMAC) using TALON Metal Affinity Resin (Clontech Laboratories Inc.) and desalted using a dialysis membrane.

### Enzyme assays

MtLAM, MsLAM, and MsAG were used as substrates for D-arabinan-degrading enzymes. The reaction products were separated using Amicon Ultra YM-10 (10 kDa cut-off, Merck Millipore, Burlington, MA, USA). The oversized fraction was loaded onto 15% polyacrylamide gel, and LAM and LM were visualized by silver-PAS staining using Sil-Best Stain-Neo (Nacalai Tesque Inc.)[47]. The filtrate was used to analyze oligosaccharides liberated by HPAEC-PAD and TLC. A CarboPac PA-1 column (4 mm internal diameter × 250 mm; Dionex Corp., Sunnyvale, CA, USA) was used for HPAEC-PAD. The fractions were eluted at a flow rate of 1.0 mL/min using the following gradient: 0–5 min, 100% eluent A (0.1 M NaOH); 5–45 min, 0–80% eluent B (0.5 M sodium acetate and 0.1 M NaOH); and 45–50 min, 100 % eluent B. For TLC analysis, the reaction products were spotted on a silica gel 60 aluminum plate (Merck Millipore) using a 7:1:2 (v/v/v) n-propanol/ethanol/water solvent mixture. The sugars were visualized by spraying the orcinol-sulfate reagent (10:1 mixture of 1% $FeCl_3$ in 10% $H_2SO_4$ and 6% ethanolic orcinol) on the plate. The plate was then dried and heated at 130 °C for approximately 1 min. Diphenyl amine-aniline-phosphoric acid reagent (1 mL of 37.5% HCl, 2 ml aniline, 10 mL of 85% $H_3PO_3$, 100 mL ethyl acetate, and 2 g diphenylamine) was used for the detection of pNP-β-D-GlcNAc hydrolysate.

The reducing power of the reaction products from the enzymatic reaction of EndoMA1 and EndoMA2 was quantified at various temperatures and pH values using MtLAM as the substrate. Under standard assay conditions, 0.50 mg/mL MtLAM in 50 mM sodium phosphate (pH 6.5) was incubated with the enzyme at 37 °C for 20 min. After mixing, the solution with an equal volume of a bicinchoninic acid solution[48], the mixture was heated at 95 °C for 15 min. After cooling for 5 min at room temperature, absorbance was measured at 560 nm. The pH profiles were measured in 50 mM sodium acetate (pH 3.5–6.0) or 50 mM sodium phosphate (pH 6.0–8.0).

The substrate specificities of EndoMA1 and EndoMA2 were analyzed by HPAEC-PAD using synthesized substrates with an acetonide tag. Enzymatic hydrolysis results in the formation of reducing end products, which were analyzed by HPLC using the p-aminobenzoic ethyl ester (ABEE) labeling[49]. The ABEE derivatives were separated using a Cosmosil SugarD column with a mobile phase of $CH_3CN$/water (70/30, v/v) at a constant flow rate (1.0 m/min) at 30 °C. The elution was monitored using a fluorescence detector (FP-202, JASCO) with excitation and emission wavelengths of 305 and 360 nm, respectively.

Transglycosylation reactions of EndoMA1 were performed using A5BT as the donor and methanol as the acceptor. A5BT (0.5 mg/mL) was incubated at 50 °C for 4 h with EndoMA1 (12 µg/mL) in 50 mM sodium phosphate buffer (pH 6.5) and 10–30% methanol. Subsequently, the reaction products were analyzed using TLC, as described above. For structural analysis, the transglycosylation product from the reaction in 30% methanol was purified using activated carbon (Autoprep Fiber; Resonac Corp., Tokyo, Japan).

The substrate specificities of ExoMA1 and ExoMA2 toward pNP-substrates were analyzed as follows. For ExoMA1, each pNP-substrate (5 mM) was incubated in the presence or absence of the enzyme in 50 mM sodium acetate (pH 5.5) at 37 °C overnight (22 h). For ExoMA2, each pNP-substrate (5 mM) was incubated in the presence or absence of the enzyme in 50 mM sodium phosphate (pH 6.0) at 37 °C overnight. The reaction products were analyzed by TLC as described above.

ExoMA1 and ExoMA2 were subjected to transglycosylation reactions using pNP-α-D-Araf and pNP-β-D-Araf as donors, respectively, and methanol as the acceptor. For ExoMA1, 5 mM pNP-α-D-Araf was incubated with the enzyme (40 µg/mL) and 10–30% methanol in 50 mM sodium acetate (pH 5.5) at 37 °C overnight. For ExoMA2, 5 mM pNP-β-D-Araf was incubated with the enzyme (13.6 µg/mL) and 10–30% methanol in 50 mM sodium phosphate buffer (pH 6.0) at 37 °C overnight. Subsequently, the reaction products were analyzed using TLC. The transglycosylation product from the reaction in 10% methanol was purified using activated carbon (Autoprep Fiber) and then subjected to structural analysis.

The pH and temperature profiles and kinetic parameters of ExoMA1 and ExoMA2 were measured by quantifying the released pNP using a UV-visible spectrometer. The absorbance at 400 nm was measured after the assay solution (25 µL) was mixed with 120 mM $Na_2CO_3$ (30 µL) to stop the reaction. The standard assay conditions for ExoMA1 were as follows: 1 mM pNP-α-D-Araf as the substrate in 50 mM sodium acetate (pH 5.5) at 37 °C. The standard assay conditions for ExoMA2 were as follows: 4 mM pNP-β-D-Araf as the substrate in 50 mM sodium phosphate (pH 6.0) at 37 °C. The pH profiles were measured in 50 mM sodium acetate (pH 4.0–6.0) or 50 mM sodium phosphate (pH 6.0–8.0).

### NMR and ESI-TOF MS analysis

The transglycosylation products, Me-α-A4B for EndoMA1 and Me-β-D-Araf for ExoMA2, were analyzed using NMR on an ECX400 spectrometer (JEOL) at 400 MHz in $D_2O$. NMR analysis of the transglycosylation products Me-α-D-Araf has been previously reported[17]. Positive ion mode ESI-TOF MS (microTOF II, Berucker Daltonics) analysis was performed on a mixture of crude products resulting from the hydrolysis process of Araf22BT with EndoMA1. The hydrolysis reaction between ExoMA2 and 20 mM pNP-β-D-Araf in $D_2O$ was monitored using NMR. A portion (650 mL of a 20 mM solution in phosphate buffer at pH 6.0 exchanged to $D_2O$ from $H_2O$ via lyophilization) of the substrate pNP-β-D-Araf was mixed with 8.2 mL of the

enzyme solution (531 μg/mL in D$_2$O). $^1$H NMR spectra of the reaction mixtures were recorded at 37 °C using an ECX400 spectrometer (JEOL) operating at 400 MHz. $^1$H NMR spectra of authentic samples, such as the initial pNP-β-D-Ara*f* and D-arabinose, reached equilibrium and were obtained without adding the enzyme.

## Crystallography

Protein samples of EndoMA1 (WT and D51N mutant), ExoMA1, and ExoMA2 used for crystallization experiments were expressed in *E. coli* and purified using IMAC, anion exchange chromatography, and SEC columns. Details of this procedure are described in the Supplementary Methods section. All proteins were crystallized using the sitting-drop vapor-diffusion method by mixing the protein and reservoir solutions (typically 0.5 μL each). The EndoMA1 D51N mutant co-crystallized with A9LT was prepared using the microseeding method by mixing reservoir, protein (32 mg/mL EndoMA1 and 10 mM A9LT), and seed solutions at a ratio of 6:4:2 and grown at 30 °C. The seed solution was prepared by crushing EndoMA1 crystals with Seed Bead (Hampton Research, Aliso Viejo, CA, USA) and diluting them 1000-fold with the reservoir solution. Crystals of the other proteins were grown at 20 °C without the microseeding method. The conditions of the protein solution, reservoir solution, and cryoprotectant are listed in Supplementary Tables 3 and 5. The crystals were cryo-cooled by dipping them in liquid nitrogen. X-ray diffraction data were collected at 100 K on the beamlines of the Photon Factory of the High Energy Accelerator Research Organization (KEK, Tsukuba, Japan). Preliminary diffraction data were collected at SPring-8 (Hyogo, Japan). The software and servers used for protein crystallography and analysis are described in Supplementary Methods.

## Mutant analysis

Site-directed mutants of EndoMA1 were constructed using PrimeSTAR Max DNA Polymerase (Takara Bio Inc.) and the primers listed in Supplementary Data 3. The mutant proteins were prepared, purified, and assayed using the same procedure as for WT EndoMA1 (recombinant protein), as described above.

## SEC-MALS/RI and SEC-SAXS of EndoMA1

SEC-MALS/RI was performed using a DAWN HELEOS II (Wyatt Technology) and an HPLC system (Alliance 2695, Waters)[50] equipped with a 2414 online differential refractometer (Waters). A KW403-4F column (Shodex) was used for SEC-MALS/RI and SEC-SAXS analyses. The measured data were analyzed using the ASTRA 6.1 software (Wyatt Technology). The SEC-SAXS experiment was conducted at the beamline BL-10C of the Photon Factory (Tsukuba, Japan)[51]. The SEC of SEC-SAXS was performed using the same column and buffer set up for SEC-MALS/RI on an HPLC system, Prominence-i (SHIMADZU). Data reduction processes were performed using the SAngler software[52], and the whole SAXS and UV-visible absorption data were analyzed using the software MOLASS[53]. The EndoMA1 apo structure was used as the reference crystal structure. The experimental conditions and analysis of the SEC-SAXS experiments are summarized in Supplementary Data 1. Details of the SEC-MALS/RI and SEC-SAXS experiments and analyses are described in Supplementary Methods.

## Electron microscopy

Purified ExoMA1 (0.1 mg/mL) in 10 mM Tris-HCl (pH 7.5) and 500 mM NaCl was applied to carbon-coated copper grids (Nisshin-EM, Tokyo, Japan) glow-discharged by HDT-400 (JEOL). The grids were negatively stained three times with 2% uranyl acetate. Electron micrographs were taken using a JEM-1400 electron microscope (JEOL) operated at 120 kV.

## Reporting summary

Further information on research design is available in the Nature Portfolio Reporting Summary linked to this article.

## Data availability

Atomic coordinates and structural factors of EndoMA1 apo, EndoMA1 + A9LT, ExoMA1, ExoMA2 + Tris, and ExoMA2 + β-D-Ara*f* have been deposited in the PDB under accession numbers 8HHV, 8IC1, 8IC8, 8IC6, and 8IC7, respectively. The draft genome of *Mi. arabinogalactanolyticum* JCM 9171 has been deposited to DDBJ under accession number SAMD00521460 (BioProject accession number, PRJDB14210; raw sequence reads in the DRA accession number, DRA0147510). Sequence Read Archive is publicly available: DRR401167. The 13 contig sequences are publicly available on GenBank at NCBI with IDs BRZC01000001–BRZC01000013. The SAXS data has been submitted to SASBDB with the deposition ID of SASDQQ8. Source data are provided with this paper.

## Materials availability

All biological materials will be distributed upon request.

## Code availability

No custom code was used in this study.

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

## Acknowledgements

We thank the staff of the Photon Factory and SPring-8 for X-ray data collection (proposal numbers 2016G083, 2019G017, 2022G034, 2014R-40, 2015R-36, 2016B2714, 2017A2714, 2017A2540, 2018A2528, 2019B2556, 2020A2553, 2022B2537, 2022PF-B014) and Dr. Ryosuke Teranishi (The University of Tokyo) for electron microscopy operation. We also thank Ms. Kayo Mimata, Mr. Kentaro Hiramatsu, Dr. Taiki Futagami (Faculty of Agriculture, Kagoshima University), Dr. Yasuo Suda and Dr. Masahiro Wakao (Graduate School of Science and Engineering, Kagoshima University) for their experimental support and helpful discussions. We thank Professor James Ketudat Cairns for the helpful discussions. This work was supported by JSPS-KAKENHI [15H02443 to S.F., K.F., and A.I.; 26660083 to S.F.; and 24380053 to S.F. and K.F.] and Mizutani Foundation for Glycoscience, and partially supported by JSPS-KAKENHI [19H00929 to S.F. and A.I.; 19K05789 to S.F.; 18K05345 to A.I.; 16H06290 to Y.I. and A.I.; 19K05816 to K.F.; JP19K06516 to N. Shimizu], the Research Support Project for Life Science and Drug Discovery [Basis for Supporting Innovative Drug Discovery and Life Science Research (BINDS)] from AMED under grant number JP22ama121001 (support number 4450 to S.F.), and the University of Tokyo Advanced Characterization Nanotechnology Platform in the Nanotechnology Platform Project sponsored by MEXT, Japan.

## Author contributions

K.F., S.F., and A.I. conceived the project and designed the experiments. M.S., N. Sawai, M.N., Y.T., A.K., S.M., N.I., K.K., and K.F. purified and biochemically characterized the enzymes. T.K., C.N., J.L., R.F., T.A., C.Y., and S.F. performed protein crystallography. A.I., K.T., and Y.I. synthesized the organic compounds and performed NMR analysis. G.A. and N. Shimizu performed SEC-MALS/RI and SEC-SAXS measurements and analysis. S.F., M.S, A.I., T.K., N. Shimizu and K.F. drafted and edited the manuscript. All authors have reviewed the data and contributed to the completion of the manuscript.

## Competing interests

The authors declare no competing interests.

## Additional information

¹Faculty of Agriculture, Kagoshima University, Kagoshima 890-0065, Japan. ²Cluster for Pioneering Research, RIKEN, Saitama 351-0198, Japan. ³Department of Biotechnology, The University of Tokyo, Bunkyo-ku, Tokyo 113-8657, Japan. ⁴Photon Factory, Institute of Materials Structure Science, High Energy Accelerator Research Organization (KEK), Tsukuba, Ibaraki 305-0801, Japan. ⁵Faculty of Pharmaceutical Sciences, Tokyo University of Science, Noda, Chiba 278-8510, Japan. ⁶School of Agriculture, Meiji University, Kawasaki, Kanagawa 214-8571, Japan. ⁷Department of Chemical Science and Engineering, Tokyo Institute of Technology, Meguro-ku, Tokyo 152-8552, Japan. ⁸Graduate School of Science, Osaka University, Osaka 560-0043, Japan. ⁹CRIIM, The University of Tokyo, Bunkyo-ku, Tokyo 113-8657, Japan. ¹⁰These authors contributed equally: Michiko Shimokawa, Akihiro Ishiwata, Toma Kashima. ✉e-mail: asfushi@mail.ecc.u-tokyo.ac.jp; k4022897@kadai.jp

