## [Peer Review File · Nature Communications]

Identification and characterization of endo- and exo-hydrolases cleaving the α - and β -D-arabinofuranosidic bonds of lipoarabinomannan and arabinogalactan of MycobacteriaREVIEWER COMMENTS

Reviewer #1 (Remarks to the Author):

This is a well-executed and well written study detailing an important enzymatic activity that could go towards Mycobacterium treatment. The authors have identified and structurally characterized three enzymes that work together to degrade lipoarabinomannan within the Mycobacterial cell wall. As my expertise is in carbohydrate-active enzymes and structural biology, though not these GH families specifically, my comments are perhaps more general. But overall, I did not see any flaws in the study execution, data or its interpretation. I have only a few comments for the authors to consider.

1. How this locus was identified as responding to MslAM was a little ambiguous to me. Clearly from the data presented the Endo and Exo enzymes are involved in degrading the target carbohydrate, but to say these are truly within a locus together with the ABC transporter – which to me and I think many folks implies there is co-regulation – requires a bit more validation. It would be most impactful to see if genes within this locus indeed turn on during exposure to MslAM (did I somehow overlook that this was done?), however I don't think this experiment is necessary or changes the main story line. Perhaps it would be good to simply remind the reader that this isn't a validated PUL yet.

2. Along the lines of my point above, does the ABC transporter bin into a particular family or have homology to existing structures? This piqued my interest.

3. I was not completely clear on the significance of the transglycosylation activity – does this just help validate the mechanism? Can the enzyme also join longer oligosaccharides?

4. The CBM finding is exciting! Is this a new family? It sounds like it has homology to members of more than one family. Can the authors expand on this a little bit?

5. The DUF4185 proteins – is the DUF4185 part of the catalytic domain and therefore a new GH family?

5. Minor - The mention of BACUNI_00161 seemed unexpected on page 7 – a bit more context for this enzyme would be appropriate at its first mention.

6. Minor - Page 7 – Tyr193 should be Tyr293 to match the Figure (or vice versa).

7. Minor - I guess labeling the bacteria as *My. smegmatis* and *Mi. arabinogalactanolyticum* is to distinguish the genus names, but I still feel this is somewhat unconventional and think using the single letter abbreviation for genus would be better.

Reviewer #2 (Remarks to the Author):

This manuscript by Shimokawa et al. presents a deep biochemical and protein structural analysis of a four-enzyme system for the hydrolysis of complex arabinan structures that decorate the surfaces of Mycobacteria relevant to human health. The identification and characterisation of two endo-arabinanases, which allow the definition of a new Glycoside Hydrolase (GH) family, represents a masterful work that builds on unresolved observations of this activity in crude culture media from ca. 50 years ago. Overall, this is a very significant advance in our understanding of mycobacterial cell wall recycling.

The manuscript is generally well-written, clear, and compelling. However, the authors should address the following minor comments to improve the presentation of the work further:

1. Abstract: The abstract should be carefully revised for grammatical accuracy ("structures serve to immune-evasive of pathogenic mycobacteria"). More importantly, the Abstract should conclude with 1-2 sentences of the potential implications and impact of the study on the field. As it is now, it reads as a simple statement of facts to emerge from the work.
2. The first two paragraphs of the Introduction are rather awkward, and currently do not read as smoothly as might be expected for a publication in this journal. I would suggest the following:
 - Revise the first sentences to something like "Arabinose is distinct among the monosaccharides, because it is widely found in nature in both the D- and L-enantiomers. L-Arabinose is a major component of plant cell walls as a component of pectin."
 - Follow this by moving current paragraph 2 up, so that the occurrence of D-arabinose in nature is described.
 - Move the sentence in the first paragraph describing the GH family distribution of the L-Araf'ases down to become the first sentence of current paragraph 3, and make any other necessary revisions to contrast the abundance of information on L-arabinose-active enzymes vs. D-arabinose-active enzymes. Delete

the sentence describing the Fru'ase activity; it is a distraction here, and can be discussed later in the context of ExoMA1 (GH172).

3. Results, first paragraph: Please make sure "EndoMA" is defined at first use.

4. Results, bottom of p.3: As this appears not to be a canonical Bacteroidetes-type PUL please include a reference to a review defining these and alternate types of PULs; e.g. Grondin et al. J Bacteriol. 2017 Jul 11;199(15):e00860-16. doi: 10.1128/JB.00860-16

5. p.5 bottom - p.6 top, also p.7: Please elaborate on whether this domain forms a new CBM family or not. If it is not, please state why not.

6. p.10, last paragraph: The first sentence seems very out of place here. The authors should elaborate on the relevance of this to their structural and mechanistic data.

7. SI: It would be helpful to place each Supplementary Figure legend immediately below the corresponding figure.

End of comments.

Reviewer #3 (Remarks to the Author):

In the paper entitled: "Identification and characterization of endo- and exo-hydrolases cleaving the α - and β -d-arabinofuranosidic bonds of lipoarabinomannan and arabinogalactan of Mycobacteria

", by Michiko Shimokawa et al., the authors present the work to identify and characterize four new glycoside hydrolases (GH) from *Mycobacterium arabinogalactanolyticum* that has the ability to degrade the d-arabinan core structure of lipoarabinomannan and arabinogalactan. These are glycan that can be found in the cell wall of acidophilic bacteria, such as *Mycobacterium tuberculosis* and *Mycobacterium leprae*.

The four d-arabinan, lipoarabinomannan and arabinogalactan active GHs from *M. arabinogalactanolyticum* that are presented in the paper are characterized intensively biochemically. And the crystal structures of three of these enzymes are determined both with and without bound glycans.

Overall the presented study is novel and includes an impressive amount of work and new results. The different types of studies included are all well planned and well performed. The results from these studies are all of high standard and well-motivated to be included in the paper.

The new findings presented in the paper are very important due to that these contribute with new information that can be used in the ongoing work worldwide to find new potent therapies for tuberculosis and other mycobacterial infections.

Major comments

This reviewer has no major comments on the paper.

Minor comments

On page 4 in the manuscript it is stated: -although EndoMA1 had a higher optimal temperature. The authors should also here state for what this optimal temperature is valid for.

On page 5 in the manuscript it is written: -ExoMA2 had an optimum pH of 6.0, The authors should also here state for what this pH optimum is valid for.

Response to specific reviewer comments

Reviewer #1

This is a well-executed and well written study detailing an important enzymatic activity that could go towards Mycobacterium treatment. The authors have identified and structurally characterized three enzymes that work together to degrade lipoarabinomannan within the Mycobacterial cell wall. As my expertise is in carbohydrate-active enzymes and structural biology, though not these GH families specifically, my comments are perhaps more general. But overall, I did not see any flaws in the study execution, data or its interpretation. I have only a few comments for the authors to consider.

Response: We thank the reviewer for the supportive comments.

1. How this locus was identified as responding to MsLAM was a little ambiguous to me. Clearly from the data presented the Endo and Exo enzymes are involved in degrading the target carbohydrate, but to say these are truly within a locus together with the ABC transporter – which to me and I think many folks implies there is co-regulation – requires a bit more validation. It would be most impactful to see if genes within this locus indeed turn on during exposure to MsLAM (did I somehow overlook that this was done?), however I don't think this experiment is necessary or changes the main story line. Perhaps it would be good to simply remind the reader that this isn't a validated PUL yet.

Response: Thank you for the very constructive comment. We agree that the experiments in the first submitted manuscript alone are ambiguous as to whether these genes work as PUL or not. Therefore, we conducted an additional experiment to measure the gene expression of the bacterial cells cultured in media containing glucose (control), D-arabinose (monosaccharide), and mycobacterial cell wall extracts (MCE). The result is presented in a new supplemental figure, Fig. S3, and in the main text (page 4). In addition to the SBP gene of the ABC transporter components, most of the enzyme genes were strongly induced by D-Ara and MCE (>10 fold compared with the Glc condition). We hope this result adequately addresses the concern of Reviewer #1.

2. Along the lines of my point above, does the ABC transporter bin into a particular family or have homology to existing structures? This piqued my interest.

Response: Thank you for the constructive comment. The ORF of the putative SBP (MIAR_33310) belongs to SBP_bac8, where many sugar transporters are classified. We also added an explanation to the main text (page 4).

3. I was not completely clear on the significance of the transglycosylation activity – does this just help validate the mechanism? Can the enzyme also join longer oligosaccharides?

Response: Considering the generally accepted reaction mechanisms of anomer-retaining and inverting GHs, the former generally exhibit transglycosylation activity, while the latter does not. Since this is in agreement with many experimental data, if the enzyme has transglycosylation activity, it is considered to be a retained enzyme. However, as raised by the Reviewer, the presence or absence of transglycosylation activity alone is not sufficient to determine the reaction mechanism. Therefore, the anomers of the reaction product by transglycosylation were determined by NMR for EndoMA1 in the new DUF4185/GH183 (Supplementary Fig. 6, formerly Supplementary Fig. 5) and ExoMA2 in GH116 (Extended Fig. 5). For GH172, other enzymes in the same family were shown to be retaining enzymes, so only the presence or absence of transglycosylation activity is shown (Fig. 3). Some of endo-type anomer-retaining glycosidases can yield long transglycosylation products when oligosaccharides are used as acceptor substrates. However, we did not try it in this study due to the limit of the synthetic oligosaccharide substrates.

4. The CBM finding is exciting! Is this a new family? It sounds like it has homology to members of more than one family. Can the authors expand on this a little bit?

Response: Thank you for the positive response. The C-terminal domain of EndoMA1 has no sequence similarity to existing CBM families. However, we consider that the creation of a new CBM family needs further examination based on concrete biochemical data. Only a crystal structure has been currently obtained in the presence of a high concentration of the ligand, and there is a possibility of crystal packing effects. We are currently verifying whether this domain is really a new CBM and hope to report on this in the future. Our opinion is added to the text (page 8).

5. The DUF4185 proteins – is the DUF4185 part of the catalytic domain and therefore a new GH family?

Response: The CAZy database created a new GH183 family for the DUF4185 enzymes (<http://www.cazy.org/GH183.html>) according to the study by Al-Jourani et al. *Nature*

Communications **14**, 1–14, 2023 (doi: 10.1038/s41467-023-37839-5). We updated the family classification of the DUF4185 enzymes in this revision.

5. Minor - The mention of BACUNI_00161 seemed unexpected on page 7 – a bit more context for this enzyme would be appropriate at its first mention.

Response: Thanks for the comment. Indeed, there was a lack of explanation where BACUNI_00161 is described. We added an explanation for BACUNI_000161 (page 8).

6. Minor - Page 7 – Tyr193 should be Tyr293 to match the Figure (or vice versa).

Response: Thanks for the comment. Tyr193 was corrected to Tyr293 (page 7).

7. Minor - I guess labeling the bacteria as *My. smegmatis* and *Mi. arabinogalactanolyticum* is to distinguish the genus names, but I still feel this is somewhat unconventional and think using the single letter abbreviation for genus would be better.

Response: We agree with the Reviewer that the labeling of *My.* and *Mi.* is not a standard one. However, the genus names of *Mycobacterium* and *Microbacterium* are very similar and should be confusing to readers. Microbiologists sometimes use this two-character labeling method for genera that can be confusing (e.g., *Bacteroides* or *Bifidobacterium*; doi: 10.1038/s42003-023-04970-4).

Reviewer #2

This manuscript by Shimokawa et al. presents a deep biochemical and protein structural analysis of a four-enzyme system for the hydrolysis of complex arabinan structures that decorate the surfaces of *Mycobacteria* relevant to human health. The identification and characterisation of two endo-arabinanases, which allow the definition of a new Glycoside Hydrolase (GH) family, represents a masterful work that builds on unresolved observations of this activity in crude culture media from ca. 50 years ago. Overall, this is a very significant advance in our understanding of mycobacterial cell wall recycling.

The manuscript is generally well-written, clear, and compelling. However, the authors should address the following minor comments to improve the presentation of the work further:

Response: We thank the reviewer for the supportive comments.

1. Abstract: The abstract should be carefully revised for grammatical accuracy ("structures serve to immune-evasive of pathogenic mycobacteria"). More importantly, the Abstract should conclude with 1-2 sentences of the potential implications and impact of the study on the field. As it is now, it reads as a simple statement of facts to emerge from the work.

Response: Thanks for the suggestion. We have rewritten the abstract. The family number (GHxxx) was also updated to GH183.

2. The first two paragraphs of the Introduction are rather awkward, and currently do not read as smoothly as might be expected for a publication in this journal. I would suggest the following:

- Revise the first sentences to something like "Arabinose is distinct among the monosaccharides, because it is widely found in nature in both the D- and L-enantiomers. L-Arabinose is a major component of plant cell walls as a component of pectin."
- Follow this by moving current paragraph 2 up, so that the occurrence of D-arabinose in nature is described.
- Move the sentence in the first paragraph describing the GH family distribution of the L-Arafases down to become the first sentence of current paragraph 3, and make any other necessary revisions to contrast the abundance of information on L-arabinose-active enzymes vs. D-arabinose-active enzymes. Delete the sentence describing the Frufase activity; it is a distraction here, and can be discussed later in the context of ExoMA1 (GH172).

Response: Thank you for your comment. The suggested reorganization has resulted in a better logical flow of the Introduction. We have revised it almost exactly as you suggested. However, the GH172 part was left in the Introduction since it is the first enzyme shown to cleave the D-Araf bond.

3. Results, first paragraph: Please make sure "EndoMA" is defined at first use.

Response: Thanks for the valuable suggestion. The etymology of EndoMA has been added (page 3).

4. Results, bottom of p.3: As this appears not to be a canonical Bacteroidetes-type PUL please include a reference to a review defining these and alternate types of PULs; e.g. Grondin et al. J Bacteriol. 2017 Jul 11;199(15):e00860-16. doi: 10.1128/JB.00860-16

Response: Thanks for the valuable suggestion that reminds us of the critical difference between the import systems of PULs in *Microbacterium* and *Bacteroides*. We have described that with the review paper reference (page 4).

5. p.5 bottom - p.6 top, also p.7: Please elaborate on whether this domain forms a new CBM family or not. If it is not, please state why not.

Response: As we responded to reviewer #1 above, we consider that additional research is still required to establish a new CBM family based on this domain until concrete biochemical evidence is presented. According to the suggestion, our opinion is added to the text (page 8).

6. p.10, last paragraph: The first sentence seems very out of place here. The authors should elaborate on the relevance of this to their structural and mechanistic data.

Response: This comment is totally correct. Thank you for pointing this out. We just reorganized that paragraph to improve the logical flow (page 11).

7. SI: It would be helpful to place each Supplementary Figure legend immediately below the corresponding figure.

Response: Thank you for pointing this out. We have corrected this.

Reviewer #3

In the paper entitled: "Identification and characterization of endo- and exo-hydrolases cleaving the α - and β -d-arabinofuranosidic bonds of lipoarabinomannan and arabinogalactan of *Mycobacteria*", by Michiko Shimokawa et al., the authors present the work to identify and characterize four new glycoside hydrolases (GH) from *Microbacterium arabinogalactanolyticum* that has the ability to degrade the d-arabinan core structure of lipoarabinomannan and arabinogalactan. These are glycan that can be found in the cell wall of acidophilic bacteria, such as *Mycobacterium tuberculosis* and *Mycobacterium leprae*.

The four D-arabinan, lipoarabinomannan and arabinogalactan active GHs from *M. arabinogalactanolyticum* that are presented in the paper are characterized intensively biochemically. And the crystal structures of three of these enzymes are determined both with

and without bound glycans.

Overall the presented study is novel and includes an impressive amount of work and new results. The different types of studies included are all well planned and well performed. The results from these studies are all of high standard and well-motivated to be included in the paper.

The new findings presented in the paper are very important due to that these contribute with new information that can be used in the ongoing work worldwide to find new potent therapies for tuberculosis and other mycobacterial infections.

Response: We thank the reviewer for the supportive comments.

Major comments

This reviewer has no major comments on the paper.

Minor comments

On page 4 in the manuscript it is stated: -although EndoMA1 had a higher optimal temperature. The authors should also here state for what this optimal temperature is valid for.

Response: Thank you for pointing this out. The word “although” was inappropriate. We also described the substrate (MtLAM) used for this assay (page 4).

On page 5 in the manuscript it is written: -ExoMA2 had an optimum pH of 6.0, The authors should also here state for what this pH optimum is valid for.

Response: Thank you for pointing this out. We described the substrate (pNP-β-D-Araf) used for the assay (page 5).

Additional note

We had a contact from a specialist of GH116 family enzymes, Prof. James Ketudat Cairns (Suranaree University of Technology). He noticed the preprint of this manuscript on Research Square (<https://doi.org/10.21203/rs.3.rs-2624731/v1>). Prof Cairns pointed out that the subfamily classification of 2' including ExoMA2 was inappropriate due to the clear activity difference from other subfamilies, especially from subfamily 2 enzymes. He also pointed out that some mammal members of GH116 on the database has smaller size, probably due to splicing variance. Therefore, Supplementary Figures 18 and 19 (the last two figures) were updated to match the specialist's comments.

Figure S18: Subfamily number 2' was changed to 4. The main text was also modified (pages 10-11).

Figure S19: Amino acid residue numbers of human GBA2 on the left were changed.